# Building wet planets through high-pressure magma–hydrogen reactions

H. W. Horn[1,4 ✉], A. Vazan[2], S. Chariton[3], V. B. Prakapenka[3] & S.-H. Shim[1 ✉]

Close-in transiting sub-Neptunes are abundant in our Galaxy[1]. Planetary interior models based on their observed radius–mass relationship suggest that sub-Neptunes contain a discernible amount of either hydrogen (dry planets) or water (wet planets) blanketing a core composed of rocks and metal[2]. Water-rich sub-Neptunes have been believed to form farther from the star and then migrate inwards to their present orbits[3]. Here we report experimental evidence of reactions between warm, dense hydrogen fluid and silicate melt that release silicon from the magma to form alloys and hydrides at high pressures. We found that oxygen liberated from the silicate melt reacts with hydrogen, producing an appreciable amount of water up to a few tens of weight per cent, which is much greater than previously predicted based on low-pressure ideal gas extrapolation[4,5]. Consequently, these reactions can generate a spectrum of water contents in hydrogen-rich planets, with the potential to reach water-rich compositions for some sub-Neptunes, implying an evolutionary relationship between hydrogen-rich and water-rich planets. Therefore, detection of a large amount of water in exoplanet atmospheres may not be the optimal evidence for planet migration in the protoplanetary disk, calling into question the assumed link between composition and planet formation location.

Water is an important building block of planets and a key ingredient of their potential habitability. The notion that water is incorporated into planetary bodies through condensation at sufficiently low temperatures in the outer protoplanetary disk seems to explain the architecture of the solar system well: the inner planets are mostly dry and rocky, sometimes with a small amount of water delivered from the outer solar system, whereas Uranus and Neptune—which are believed to be ice giants—exist outside the snow line[6]. However, some nebular hydrogen-rich gas was possibly captured and stored as $H_2O$ in the deep interior of Earth after reacting with silicates[7–9].

NASA's Kepler mission has discovered many of close-in transiting exoplanets between Earth-like and Neptune-like in size (1–4 Earth radius, $R_E$) and density[10]. Planets with radii of 2–4$R_E$ are typically modelled with rocky cores and envelopes dominated by either hydrogen or water, making their composition and formation ambiguous (where 'core' refers to the non-volatile interior, as commonly defined in exoplanet literature). A drop in relative frequency of occurrence among such planets was identified at 1.5–2$R_E$ (that is, the radius valley)[1], which seems to divide planets into two groups: rocky + metallic Earth-like density with tiny or no atmosphere at 1–1.5$R_E$ (that is, super-Earths), and rocky + metallic core with thick atmosphere at 2–4$R_E$ (that is, sub-Neptunes). Models involving massive gas loss can explain the radius valley well[11,12], which seems to support hydrogen-dominated atmospheres, whereas other models explain the radius valley by two separate populations of dry and water-rich planets[13,14]. Moreover, some systems may host water-rich sub-Neptunes[15–17]. An important

question for water-rich sub-Neptune models is how planets with a substantial water mass fraction can exist in close-in orbits if water mainly condensed and was incorporated into planets in the outer part of the systems during their formation[3].

In classical models of sub-Neptunes with hydrogen-rich envelopes, hydrogen is typically assumed not to react with silicates and metals in the core, although physical mixing with silicate magma has been considered[18,19]. At low pressures and high temperatures, hydrogen can reduce cations in silicates and oxides to metals[20], releasing oxygen from melts and producing water through hydrogen oxidation. Experimental data show that hydrogen can reduce $Fe^{2+}$ to $Fe^0$ in molten silicates, generating water at low pressure[7,21,22], and this trend extends to higher pressures[23]. However, in protoplanetary disks, most iron is metallic, with only limited $Fe^{2+}$ and $Fe^{3+}$ available. Recent experiments show that Mg remains oxidized in dense hydrogen, but small amounts of $MgH_2$ can form, releasing oxygen for water production[24]. Thermodynamic models based on low-pressure data suggest that hydrogen cannot reduce a substantial amount of $Si^{4+}$ in silicates to produce water[5,7]. A modelling study indicates that enhanced Si partitioning into metal under reducing conditions could yield Earth-like water abundances (about $10^{-2}$ wt%) (ref. 9). However, experiments show that $SiO_2$ can dissolve in hydrogen at 2–15 GPa and 1,400–1,700 K, with Si–O bonds breaking to form Si–H, suggesting that pressure may promote hydrogen–silicate reactions[25,26].

If the mass–radius of sub-Neptunes is modelled with a rock + metal core with a hydrogen-dominated envelope, the pressure at the

[1]School of Earth and Space Exploration, Arizona State University, Tempe, AZ, USA. [2]Astrophysics Research Center, Department of Natural Sciences, Open University of Israel, Ra'anana, Israel. [3]Center for Advanced Radiation Sources, The University of Chicago, Chicago, IL, USA. [4]Present address: Lawrence Livermore National Laboratory, Livermore, CA, USA. ✉e-mail: hallensu@asu.edu; sshim5@asu.edu

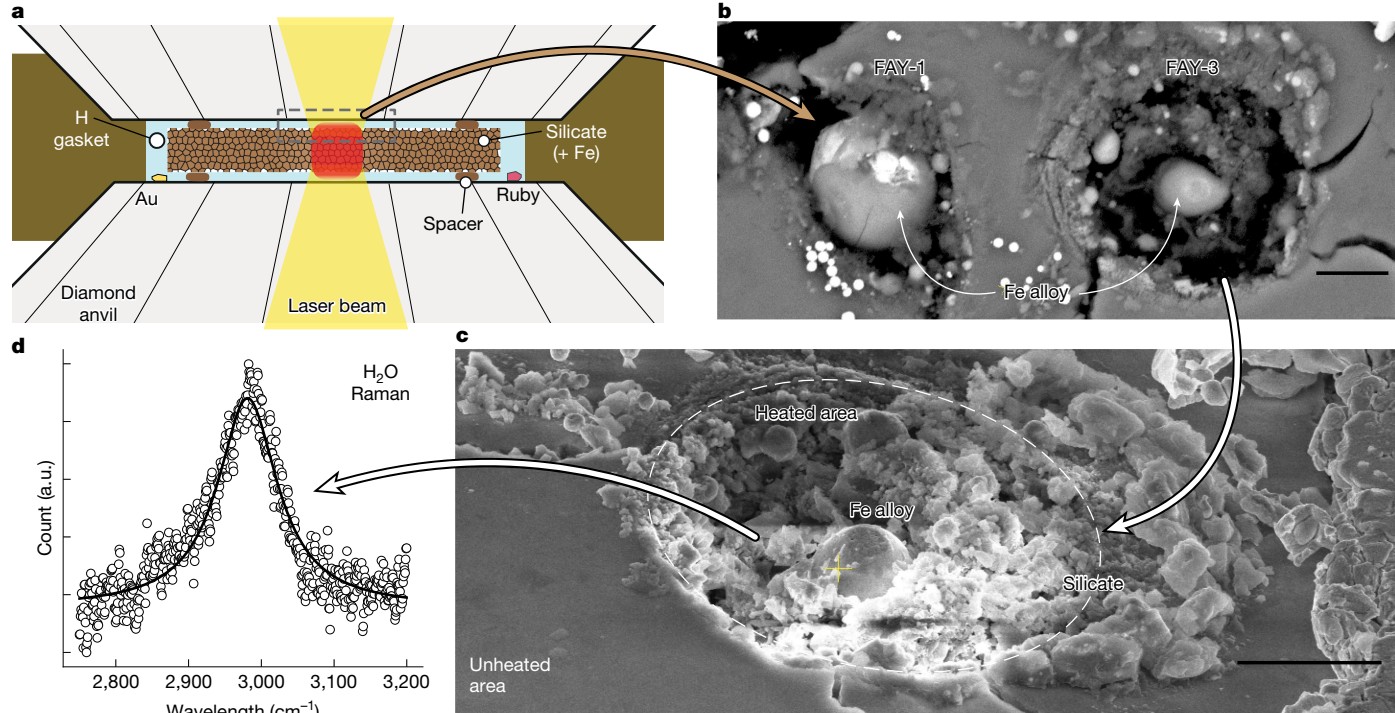

**Fig. 1 | Laser-heated diamond-anvil cell experiments on silicate melts in a hydrogen medium. a**, Schematic of the experimental setup. The spacers (small single grains of the same material from which the foil was made) separate the sample foil from diamond anvils, which allow hydrogen to surround the sample. During laser heating (the red area at the centre) of the silicate sample, hydrogen penetrated the grain boundaries of the sample foil and immediately above and below were heated by thermal conduction. **b**, An SEM image of two laser-heated areas of a fayalite sample. FAY-1 was heated at 6 GPa and 3,017 K and FAY-3 was heated at 11 GPa and 2,898 K. The spheres at the centre of the heated areas are Fe-rich alloys formed by hydrogen–silicate reaction. **c**, An SEM image of FAY-3 from an angle for a wider area. **d**, Raman-active OH vibration from $H_2O$ ice after heating silica + Fe metal in a hydrogen medium at 14 GPa. A full two-dimensional Raman map of the heated area is shown in Extended Data Fig. 3c. Scale bars, 5 μm (**b**,**c**).

core-envelope boundary (CEB) can exceed a few GPa (ref. 27). This blanketing envelope can also decrease heat loss such that silicates and metals of the core remain molten for billions of years (ref. 28). Under these conditions, hydrogen is a dense fluid. Therefore, it is essential to understand the behaviour of the hydrogen–silicate system at the high pressures and temperatures ($P$–$T$) that exist at such a CEB. However, to our knowledge, these data do not exist. In this study, we combined pulsed laser heating[29] with the diamond-anvil cell to mitigate diamond embrittlement, which has been the main problem in melting silicate and metal in a pure hydrogen medium to acquire key data for understanding possible hydrogen–silicate reactions at the $P$–$T$ conditions expected at the CEB in sub-Neptunes (Fig. 1, Extended Data Figs. 1 and 2, Extended Data Table 1 and Supplementary Discussion 1).

## High-pressure magma–hydrogen reactions

A starting mixture of San Carlos olivine, $(Mg_{0.9}Fe_{0.1})_2SiO_4$, and iron metal was melted in a hydrogen medium at 8 GPa (Extended Data Table 1, run SCO-3). After heating, high-pressure X-ray diffraction (XRD) patterns showed no peaks from silicates, including olivine (Fig. 2a). Instead, diffraction lines of B2 $Fe_{1-y}Si_y$ were observed, showing that $Si^{4+}$ in silicate melt was reduced to $Si^0$ and alloyed with Fe metal. After heating, Mg remains in MgO periclase. The unit-cell volume of MgO after decompression to 1 bar indicates no detectable $Fe^{2+}$ in the phase. Therefore, all $Fe^{2+}$ originally in the silicate is also reduced to $Fe^0$ metal, implying the following reaction:

$$(Mg_{0.9}Fe_{0.1})_2SiO_4 + 0.8Fe + 2.2H_2 \rightarrow FeSi + 1.8MgO + 2.2H_2O. \quad (1)$$

In our experiments, 20 wt% of Fe metal was mixed with olivine and a smaller amount of Fe metal was formed from the reduction of $Fe^{2+}$ in olivine. If all the Fe metal is consumed to form FeSi (that is, $y = 0.5$ in $Fe_{1-y}Si_y$), a maximum of 86% of the $Si^{4+}$ in olivine can be reduced (Supplementary Code 1). However, the measured unit-cell volume of the Fe–Si alloy indicates $y = 0.27$ in $Fe_{1-y}Si_y$ (Supplementary Discussion 2–8). Therefore, only 32% of $Si^{4+}$ in olivine was reduced to form an alloy with Fe metal. Furthermore, $FeH_x$ alloys in the face-centred cubic (fcc) and double hexagonal close-packed (dhcp) structures ($x = 0.22$–0.4) were found after heating, thus decreasing the amount of Fe metal available to form Fe–Si alloys. The observation of the complete disappearance of silicates after melting in dense hydrogen fluid requires an additional process to remove $Si^{4+}$ from silicates.

Previous experiments at similar pressures but lower temperatures[25,26] showed the release of $Si^{4+}$ from silicate and dissolution in dense fluid hydrogen as $SiH_4$:

$$Mg_2SiO_4 + 4H_2 \rightarrow SiH_4 + 2MgO + 2H_2O. \quad (2)$$

We detected the same Si–H bond vibration in melted areas in our experiment (Extended Data Fig. 4 and Supplementary Discussion 9). Both the reduction and hydride formation reactions (reactions 1 and 2, respectively) involve the release of O, which can then react with H to form $H_2O$. The OH vibration from $H_2O$ was unambiguously detected in melted areas with Raman spectroscopy (Fig. 1d and Extended Data Fig. 3). The same behaviour was also observed at pressures up to 22 GPa in multiple experimental runs (Extended Data Fig. 5 and Supplementary Discussion 2 and 3).

When olivine + Fe was heated at 42 GPa to temperatures below the melting temperature[30], olivine transformed to bridgmanite (bdm) and

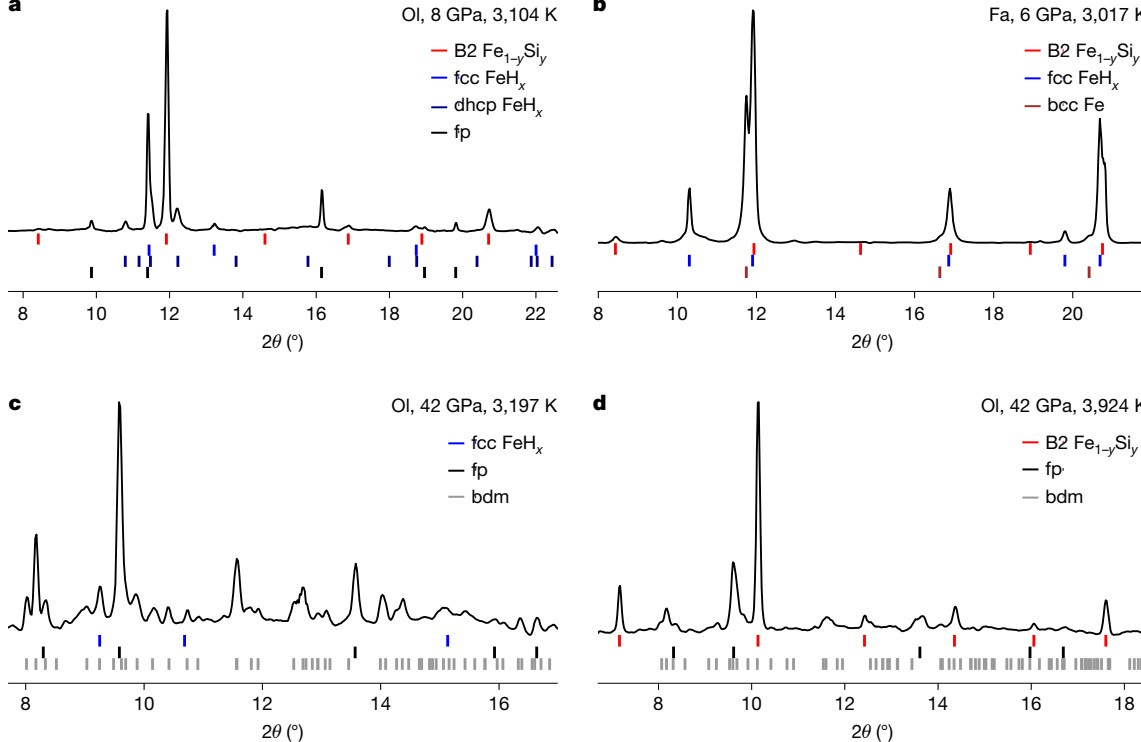

**Fig. 2 | XRD patterns after heating silicate samples in a hydrogen medium.**
**a**, In run SCO-3, olivine breaks down to Fe metal alloys and MgO. **b**, In run FAY-1, fayalite breaks down with Fe and Si present as metal alloys. A very small amount of body-centred cubic (bcc) Fe may exist. **c**, Heating to temperatures below melting (2,725–3,197 K), bridgmanite (bdm) and ferropericlase (fp) appear at 42 GPa (SCO-13a). **d**, When the bdm + fp were melted (3,352–3,924 K at 42 GPa), bdm mostly breaks down and B2 $Fe_{1-y}Si_y$ appears (run SCO-13b). X-ray energy is 30 keV for **b** and 37 keV for **a**, **c** and **d**. The ticks below the diffraction patterns are the peak positions of the observed phases. fcc, face-centred cubic; dhcp, double hexagonal close-packed; Ol, San Carlos olivine starting material; Fa, fayalite starting material.

ferropericlase (fp) (Fig. 2c). We further heated the synthesized bdm + fp above the melting temperature of bdm. Most of the bdm dissociates on melting in a hydrogen medium (Fig. 2d). The $SiO_2$ component reacts with H and Fe to form B2 $Fe_{1-y}Si_y$ (Extended Data Fig. 3a,b). The volume measured for quenched fp suggests the complete reduction of $Fe^{2+}$ to $Fe^0$ (Supplementary Discussion 4).

With fayalite ($Fe_2SiO_4$) and $SiO_2$ + Fe metal, we observed consistent behaviours, including formation of $H_2O$ (Fig. 1d), $SiH_4$ (Extended Data Fig. 4), B2 $Fe_{1-y}Si_y$ (Extended Data Fig. 6) and $FeH_x$ (Fig. 2b) (Extended Data Figs. 7–9; Supplementary Discussion 5). $SiH_4$ was detected in runs in which fayalite was melted in an Ar medium with 50% $H_2$ (Supplementary Discussion 9). In this case, the relative content of $H_2O$ (formed from the reaction) compared with $H_2$ is much higher because of the decreased initial concentration of H and the increased $H_2O$ production from the $Fe^{2+}$ in the silicate, resulting in far more oxidizing conditions. In $SiH_4$, Si remains oxidized and, therefore, water production can continue as the water concentration rises through the formation of $SiH_4$ rather than Fe–Si metal (equation (2)).

## Implications of reaction for planets

In a sub-Neptune planet with a rocky core and a substantial amount of hydrogen, the CEB is the most likely region to experience reactions between dense fluid hydrogen and silicate melt. The hydrogen–magma boundaries of these planets in the mass range of 3–10$M_E$ (where $M_E$ is mass of the Earth) with 2–20 wt% of H + He atmosphere are expected to reach $P$–$T$ conditions similar to our experimental conditions for water production (Supplementary Discussion 10). According to our calculation in ref. [28], for a 5$M_E$ rocky planet with a 5 wt% $H_2$ envelope, the temperature of the core remains sufficiently high to maintain the molten state of silicates for billions of years (Extended

Data Fig. 2), which could, in turn, make the continuation of the observed reactions possible for billions of years. Moreover, because pressure greatly enhances $H_2$ solubility in magma[18] and can result in miscibility between hydrogen fluid and oxide melts[24], hydrogen can reach greater depths below the CEB. Vigorous convection in the molten core could also transport hydrogen further to greater depths. For all of these reasons, the reactions discussed can continue into the deep interior.

We combined equations (1) and (2) with constraints from experimental setups and observations to estimate the quantities of reactants and reaction products for olivine reacting with hydrogen in our experiments (Supplementary Discussion 11). Considering uncertainties (Supplementary Code 1), we calculated that before reacting, the sample volume that would be heated consisted of 4.5–5.7 wt% $H_2$, about 76 wt% silicate and about 19 wt% Fe metal. It shows that 18.1(5) wt% $H_2O$ was produced. This composition lies within the range considered for sub-Neptunes[2]. The complete disappearance of silicate (and therefore release of all of the Si) at lower pressures suggests silicate-undersaturated conditions for the hydrogen-to-silicate mass ratio of our olivine + Fe metal experiment, which was estimated to be 0.06–0.08. For a sub-Neptune with 4 wt% $H_2$ and an Earth-like core composition, the hydrogen-to-silicate mass ratio is 0.06. Therefore, sub-Neptunes with a few wt% $H_2$ could provide a silicate-undersaturated condition with respect to hydrogen. At >25 GPa, a trace amount of bdm (silicate) remained in the reacted area (Fig. 2d), which could result from the reverse reaction and, therefore, suggest a possible equilibrium. If this is the case, even under equilibrium conditions, the amount of Si in the silicate melt consumed by the reaction is very large, and the $H_2O$ produced can amount to tens of weight per cent (Supplementary Code 1).

A theoretical study considering the formation of $SiH_4$ at hydrogen–magma boundaries[4] predicted $X(H_2O) \approx 0.2$ (where $X$ is a mole fraction

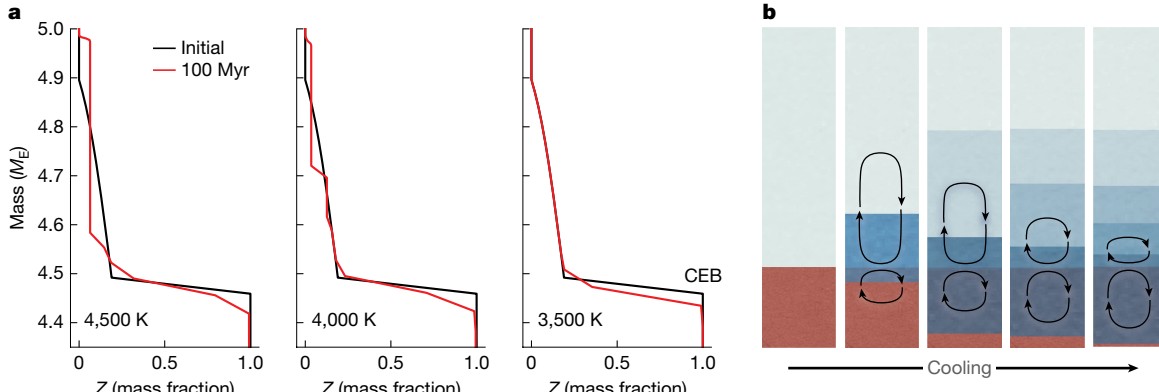

**Fig. 3 | Water redistribution in a 5$M_E$ planet with 10 wt% envelope made of H$_2$ and He. a**, Water mass fraction ($Z$) for an initial state distribution (black) and after 100 Myr of evolution (red). Mixing efficiency decreases as the planet cools (left to right). **b**, The schematic shows the time progression of the mixing scale near the reaction zone.

in the envelope) of water production along with $X(SiH_4) \approx 0.04$, and $X(SiO) \approx 0.08$ in a 4$M_E$ sub-Neptune with 2.5 wt% H$_2$ at about 10 GPa and 5,000 K. Another theoretical study[5] considering the reduction of Si predicted $X(H_2O) \approx 10^{-4}–10^{-3}$ of water production for a 4$M_E$ sub-Neptune with a CEB temperature of 4,500 K. At similar $P$–$T$ conditions, however, our experimental observations indicate much more efficient endogenic water production of $X(H_2O) = 0.38–0.56$, which is 2–3,000 times higher. Although the modelling studies used extrapolation of assumed ideal gas behaviour of H$_2$ at much lower pressure, our experiments were conducted at the $P$–$T$ conditions for dense fluid hydrogen, directly relevant to the conditions and states expected at the CEBs of sub-Neptunes. A comparison of our results with refs. 4,5 highlights the notable pressure effects on water production.

At a planetary scale, the extent of endogenic water production can be affected by the activity of H$_2$ and H$_2$O in the system. Under a hydrogen-dominated envelope lacking H$_2$O, strongly reducing conditions promote the reduction of Si as a key pathway for endogenic water production. As more water is produced (and therefore the activity of H$_2$ decreases), the reaction zone becomes less reducing. As the reaction progresses, if the water concentration reaches a point at which the conditions become sufficiently oxidizing, the reduction of Si could stop. In this case, SiH$_4$ would become a dominant Si-bearing product of the hydrogen–magma reaction because it does not require the reduction of Si$^{4+}$. Because of this, SiH$_4$ formation can occur at more oxidizing conditions than Fe–Si alloys, as shown in our experiments (Supplementary Discussion 9), prolonging endogenic water production to more oxidizing conditions.

The redox state of the reaction zone can also be affected by the dynamics of the region. In the absence of mixing and transport, the fraction of water near the reaction zone will increase, which will slow down and ultimately stop the water-producing reaction when the water concentration exceeds a critical value (which is not well constrained, but we estimate to be $X(H_2O) \geq 38–56$ mol%; Supplementary Discussion 11 and Supplementary Code 1). However, the interiors of sub-Neptunes by and large are convective, and therefore efficient mixing and transport can reduce the concentration of H$_2$O in the reaction zone. In Fig. 3a,b, we show an example of the efficiency of water-mixing over 100 million years in a 5$M_E$ rocky planet with a 10% gas (H, He) envelope. For CEB temperatures of ≥4,500 K, water is almost completely mixed through convection in the envelope. The mixing efficiency decreases as the planet cools. Because water production continues up to $X(H_2O) > 0.3$, we anticipate that convective mixing may stop before water production has ended. Consequently, the deep envelope might be more water-rich than the outer envelope in these planets at CEB temperatures lower than about 3,500 K.

Hydrides also form in the reactions discussed here and in other experiments[24–26]. Only a small amount of MgH$_2$ was reported to be produced, and its formation is limited to temperatures much higher than 3,000 K (Supplementary Discussion 13). Estimates from our experiments (Supplementary Discussion 11) show 6–23(5) mol% SiH$_4$ can be produced, and, therefore, it is important to consider. Although no data exist directly at the $P$–$T$ conditions of our study, existing data indicate that the density of SiH$_4$ is 20–30% lower than that of H$_2$O at 300 K and relevant pressures (Supplementary Discussion 14). Therefore, the upward transport of H$_2$O and SiH$_4$ away from the reaction zone could be more efficient than predicted by our model.

It is essential to note that at the $P$–$T$ conditions we consider here, H$_2$ and H$_2$O are completely miscible and form a single fluid[31], whereas our dynamic model includes only convective mixing of two separate fluids of H$_2$ and H$_2$O. Similar to H$_2$O and H$_2$, existing data also support the miscibility between SiH$_4$ and H$_2$ at the conditions of the CEB. The miscibility will decrease the density of the fluid compared with pure H$_2$O, and the hydrogen contained in the dense fluid layer will still react with the magma ocean (Supplementary Discussion 14).

The increasing solubility of water in silicates at higher pressures (that is, ingassing)[32,33] can reduce its activity at the CEB and thus prolong the hydrogen–silicate reactions. Furthermore, the dissolved water decreases both the melting temperature and viscosity of the magma, prolonging the molten state of the interior and promoting efficient convective mixing of materials to the deeper undersaturated depths of the core. Therefore, strong ingassing of water (and hydrogen) to the core can also enable more water production.

The amount of endogenic water produced can vary depending on the properties and configurations of different planets (Supplementary Discussion 12). The Mg:Si ratio can directly affect the amount of water produced, as Si-involved reactions with hydrogen probably dominate endogenic water production compared with Mg and Fe (Supplementary Discussion 13). For example, for a planet with an Earth-like metal-to-silicate mass ratio but the Mg:Si ratio reduced from 2 to 0.5 (more Si), if all the silicate reacts with hydrogen (>3 wt% of H$_2$), the amount of water produced will increase from 16 wt% to 29 wt% (Supplementary Code 3). The Mg:Si ratio will also affect the viscosity of the magma and, therefore, affect the ingassing and mixing of volatile species[34]. Therefore, the large variation in Mg:Si ratios observed in exoplanetary systems[35] will result in variation in endogenic water production among planets with rocky interiors and hydrogen-dominated atmospheres.

The proposed atmospheric loss of the hydrogen-rich envelope of a sub-Neptune[11,12] may occur concurrently with the endogenic production of water. The timing and duration of these two processes are

important factors in the amount of water produced. Atmospheric loss also plays a part in the rate of heat loss, which, in turn, influences the duration for which the water-producing reactions can persist because they predominantly occur in molten silicates and metal alloys.

If a sub-Neptune undergoes massive gas loss while water is retained because of its high mean molecular mass[22,36], it is feasible that the limited water production that occurred concurrently with gas loss would result in a rocky planet with surface water. For example, for a $5M_E$ sub-Neptune, if only the outer 5% of the rocky core reacts with hydrogen, 2–4 wt% $H_2O$ would be produced (Supplementary Code 2). At the same time, large amounts of water can also be stored in the core of sub-Neptunes because pressure increases the solubility of water in magma[32,33]. Therefore, even if surface or atmospheric water is lost during gas loss, the water stored in the core can contribute substantially to the formation of a secondary atmosphere and hydrosphere when the interior cools and solidifies, in which the solubility of water is much lower[22,37,38]. Moreover, because higher pressures markedly enhance water-producing reactions, super-Earths converted from hydrogen-rich sub-Neptunes may contain much more water than smaller rocky planets such as Earth.

Another key implication is that hydrogen-rich and water-rich sub-Neptunes do not necessarily form through different processes[39,40]. Instead, the reaction we report here suggests that these planet types may be fundamentally related: hydrogen-rich sub-Neptunes could be the precursors of water-rich sub-Neptunes and super-Earths. If an excess, unreacted $H_2$ atmosphere can be retained, sub-Neptunes with an $H_2$-rich atmosphere covering an $H_2O$-rich layer above the core (that is, hycean worlds) may be quite common[41].

In conventional planet-formation theory, water-rich planets are believed to form outside the snow line. The observation of water-rich sub-Neptunes[15–17,42,43] in close orbits raises important questions about how they can form. Models have been proposed to explain close-in transiting water-rich sub-Neptunes, such as migration of water worlds after their formation outside the snowline[3,40,44]. Endogenic water production through hydrogen–magma reactions observed in our experiments provides a straightforward process to build water-rich sub-Neptunes inside the snow line: close-in orbiting sub-Neptunes can be $H_2O$-rich planets, as the hydrogen–silicate reaction can convert the H-dominated atmosphere to $H_2O$ from the inside out, almost independent of their radial distance from the stars. Water production is feasible and expected in the large population of $3–10M_E$ planets that were formed with substantial (>2%) gas envelopes. On planets with a hotter CEB, vigorous convection and sustained water production are expected to persist longer over their thermal evolution. Moreover, if mixing of water into the core is efficient, the production of water can persist even when convection in the water-polluted envelope becomes less efficient. This result has fundamental implications for planet formation and evolution theories. Based on experimental results, we suggest that planets formed from dry materials can become water-rich (tens of wt% water) planets without direct accretion of water ice. Consequently, detection of a large amount of water in exoplanet atmospheres might not be the optimal evidence for planet migration in the protoplanetary disk. Our new experimental findings challenge the assumed link between planet formation location and composition.

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

## Methods

### Sample materials and preparation

We studied three starting materials: (1) San Carlos olivine, $(Mg_{0.9}Fe_{0.1})_2SiO_4$, (2) natural fayalite, $Fe_2SiO_4$ (Smithsonian: R-3517-00: Rockport) and (3) silica (Alfa Aesar 99.995% purity). For laser coupling, silica was mixed with Fe metal powder (Aldrich 99.9%+ purity). Although San Carlos olivine contains enough Fe for laser coupling, to further improve the coupling, we mixed San Carlos olivine with Fe metal powder (20 wt%). Powders were ground and mixed in an alumina mortar, then cold-pressed into foils with a thickness of approximately 10 μm. Rhenium gaskets were indented by diamond anvils with 200 μm diameter culets and then drilled with 125 μm diameter holes. The rhenium gaskets were then coated with about 800 Å of gold to inhibit hydrogen diffusion into the gasket material. The gaskets were placed back onto the diamond culets, followed by the sample foils and spacers (of the same material as the sample foil), and gold and ruby grains for pressure calibration. The cells were then loaded with 1,300–1,500 bars of pure $H_2$ gas in a Sanchez GLS 1500 gas loading system and then compressed to the target pressures (measured using ruby fluorescence[45]) at 300 K before synchrotron laser-heating experiments. A fayalite sample was loaded with a 50:50 gas mixture of $H_2 + Ar$ to examine the impact of a lower concentration of $H_2$.

### Synchrotron experiments

In situ XRD images of the samples in the laser-heated diamond-anvil cell (LHDAC) were collected at the 13-IDD beamline of the GeoSoilEnviroConsortium for Advanced Radiation Sources sector at the Advanced Photon Source (APS) synchrotron facility. Near-infrared laser beams of wavelength 1,064 nm and monochromatic X-ray beams of wavelength 0.4133 Å or 0.3344 Å were coaxially aligned and focused on the sample in the LHDAC[46]. Standard continuous laser heating of H-loaded samples in the DAC results in an extremely mobile and diffusive hydrogen fluid that can penetrate diamond anvils, leading to diamond embrittlement and failure of the anvils[47]. To enable melting of silicates in a hydrogen medium, a pulsed laser-heating system was used to mitigate the amount of hydrogen diffusion into the anvils and the gasket material[29]. Each pulsed heating event consisted of $10^5$ pulses at 10 kHz and 20 streak spectroradiometry measurements. The pulse width was 1 μs. Therefore, an event with $10^5$ pulses gives a total heating time of 0.1 s. The X-ray spot size is $3 \times 4$ μm$^2$, and the laser-heating spot is a circle with a diameter of approximately 15 μm. The laser pulses were synchronized with the synchrotron X-ray detector such that diffraction measurements could take place only when the sample reached the highest temperature during heating. The small X-ray beam size and large laser-heating spot size help mitigate the effects of the radial thermal gradients in the high-temperature diffraction patterns. A previous study[46] showed that the laser-heating system provides a flat top laser beam intensity profile, which further reduces the radial thermal gradients in the hot spot. We also conducted two-dimensional XRD mapping after temperature quench of the samples in LHDAC at high pressures to monitor the possible effects of thermal gradients during laser heating. In the in situ high-temperature XRD patterns, it was difficult to distinguish broad diffuse scattering features from the melt because of the short duration of the measurements.

Although the heating duration is short, it was found that hydrogen is extremely reactive with molten silicates at high temperatures, overcoming the limited heating exposure time. Moreover, sample foils cold-compressed from powder permit hydrogen gas and fluid to percolate and surround individual grains (less than 1 μm in size). This creates a large surface area of the silicate exposed to hydrogen and, therefore, facilitates a fast reaction[48]. A recent study obtained consistent results between short pulse heating and continuous heating under hydrogen-rich conditions in diamond-anvil cells[49].

Thermal emission spectra from both sides of the sample in LHDAC were fitted to the grey-body equation to estimate the temperatures[46].

Temperatures are assigned from an average of temperatures recorded from 20 measurements (10 each upstream and downstream) over a heating event (Extended Data Table 1). Temperature uncertainty was calculated from the standard deviations ($1\sigma$) of these measurements. If the standard deviation is smaller than 100 K, from intrinsic uncertainties in the spectroradiometry method, we assigned 100 K for the uncertainty[50].

A Dectris Pilatus 1M CdTe detector was used to collect two-dimensional diffraction images. The diffraction images are integrated into one-dimensional diffraction patterns using the DIOPTAS package[51]. Diffraction patterns of $CeO_2$ and $LaB_6$ were measured for the correction of detector tilt and the determination of the sample-to-detector distance. Unit-cell parameter fitting was conducted by fitting the diffraction peaks with pseudo-Voigt profile functions in the PeakPo package[52]. Pressure was calculated from the unit-cell volume of a gold grain at the edge of the sample chamber using the equation of state of gold[53] before and after heating. A gold grain was placed away from the sample rather than mixing with it to prevent reactions or alloying with the sample material, and thus, pressure could not be measured at high temperatures during laser heating. A previous study[54] showed that the thermal pressure in a liquid Ar medium at temperatures of 1,000–4,000 K is 0.5–2.5 GPa in LHDAC. All of our experiments exceed the melting temperature of hydrogen. Therefore, thermal pressure should be similar to the above estimation in our experiments, and we assign a pressure uncertainty of 10% for high-temperature data points[23]. We note that this method does not introduce a severe error for this study, which is to explore the hydrogen–silicate reactions at high pressures.

### Raman spectroscopy

Raman measurements were conducted using the Raman spectroscopy system at GeoSoilEnviroConsortium for Advanced Radiation Sources[55] for the identification of O–H and Si–H vibrations after heating. Raman scattering of the sample in a diamond-anvil cell was excited by a monochromatic 532-nm beam from a Coherent VERDI V2 laser. Raman spectra were collected over a wide range of wavenumbers (1,400–4,500 cm$^{-1}$) using a Princeton Instruments Acton Series SP-2560 spectrograph and PIXIS100 detector.

### Modelling for the dynamics of the interior

The thermal evolution model is calculated for the entire interior (from centre to surface) on one mass grid, with no distinction between core and envelope (Fig. 3). The model is one-dimensional and solves the standard interior structure and evolution equations, which allow for heat transport by convection, radiation and conduction, depending on local conditions over time (equations can be found in ref. 56). The basic set of parameters for rocky planets with gas envelopes is adopted from ref. 28. The redistribution of composition by convective mixing, in which the convection criterion is fulfilled, is calculated as in ref. 56.

The input equations of state are from ref. 57 for hydrogen and helium, and an improved version of ref. 58 for water and silica as representatives of volatiles and refractories, respectively. In our experiments, we found that multiples of components, such as $SiH_4$ and $MgH_2$, apart from $H_2O$, can exist in $H_2$ envelopes. In this system, semi-convection (double diffusive convection) might develop under certain conditions (Supplementary Discussion 14), which could limit the mixing efficiency and terminate water production earlier than our models suggest. Irradiation by the parent star is included as a temperature boundary for a plane-parallel grey atmosphere with an optical depth of 1. The radiative opacity is that of a grain-free solar metallicity atmosphere[59].

### Data availability

The experimental data that support the findings of this study are available from the corresponding authors upon reasonable request.

The XRD and Raman data are available at Zenodo (https://doi.org/10.5281/zenodo.15586691)[60]. An overview of the data is also included there. Source data are provided with this paper.

## Code availability

Supplementary codes used in this study can be found at Zenodo[61] (https://doi.org/10.5281/zenodo.15678598).

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

**Acknowledgements** We thank Y. Fei and G. Huss for their comments and questions, which improved the quality of this paper. S.-H.S. and H.W.H. were supported by the National Science Foundation (NSF) grants AST-2108129, AST-2406790 and EAR-1921298. Portions of this work were performed at GeoSoilEnviroCARS (The University of Chicago, Sector 13), Advanced Photon Source, Argonne National Laboratory. GeoSoilEnviroCARS is supported by the NSF, Earth Sciences (EAR-1634415). This research used resources of the Advanced Photon Source, a US DOE Office of Science User Facility operated for the DOE Office of Science by the Argonne National Laboratory under contract no. DE-AC02-06CH11357. We acknowledge the use of facilities in the Eyring Materials Center at ASU. Part of this work was performed under the auspices of the US DOE by Lawrence Livermore National Laboratory under contract DE-AC52-07NA27344. The opinions are those of the authors and do not necessarily represent the opinions of LLNL, LLNS, DOE, NNSA or the US government. The US government, and the publisher, by accepting the article for publication, acknowledge that the US government retains a non-exclusive, paid-up, irrevocable, worldwide licence to publish or reproduce the published form of this manuscript, or allow others to do so, for US government purposes (release authorization no. LLNL-JRNL-872168). A.V. acknowledges support by ISF grants 770/21 and 773/21.

**Author contributions** H.W.H. and S.-H.S. conceived the project; H.W.H., S.C., V.B.P. and S.-H.S. conducted the synchrotron experiments; A.V. conducted sub-Neptune modelling; H.W.H. and S.-H.S. analysed the results; and H.W.H., S.-H.S. and A.V. wrote the paper. All authors reviewed the paper.

**Competing interests** The authors declare no competing interests.

**Additional information**
**Correspondence and requests for materials** should be addressed to H. W. Horn or S.-H. Shim.

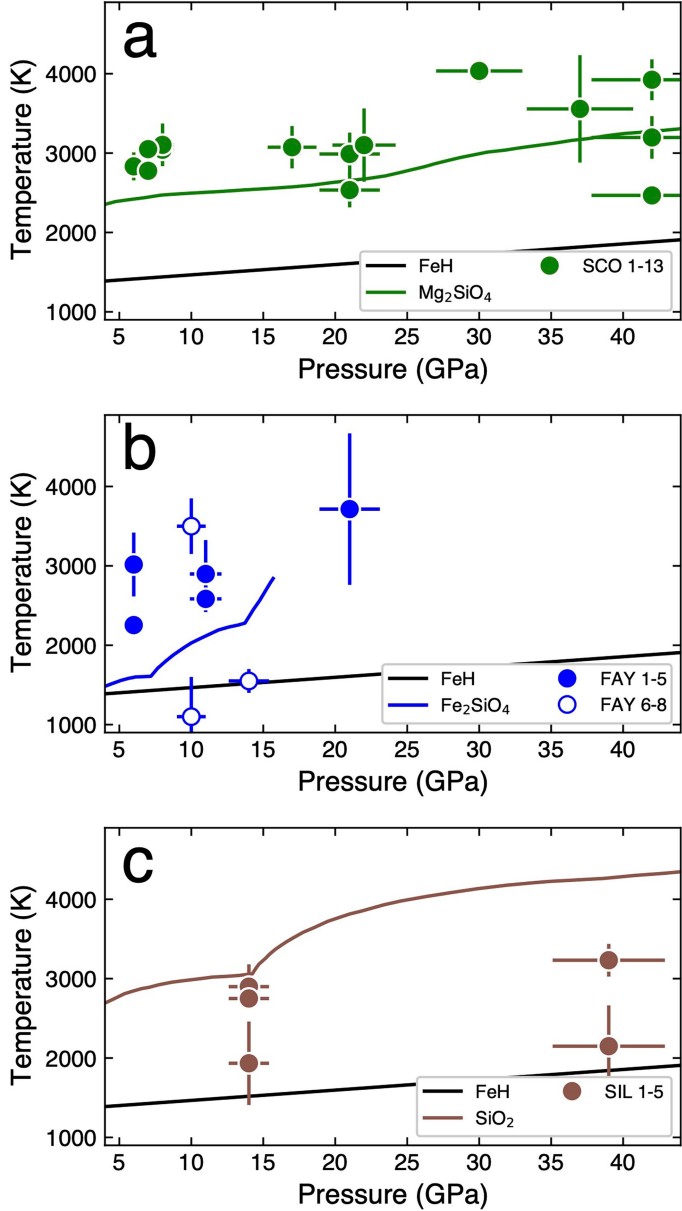

**Extended Data Fig. 1 | Pressure-temperature conditions of the experimental runs. a**, San Carlos olivine, **b**, fayalite, and **c**, silica starting materials in this study (Extended Data Table 1). For fayalite, three experimental runs were conducted with a 50% Ar + 50% $H_2$ medium (open circles). Melting curves for the relevant phases are shown: $FeH_x$ (ref. 62); $Mg_2SiO_4$ (ref. 63); $Fe_2SiO_4$ (ref. 64); $SiO_2$ (ref. 65). All the experiments were conducted above the melting temperature of hydrogen[66]. While olivine and fayalite couple with laser beams sufficiently well for melting, lack of Fe in silica makes it difficult to heat above melting as shown in **c**.

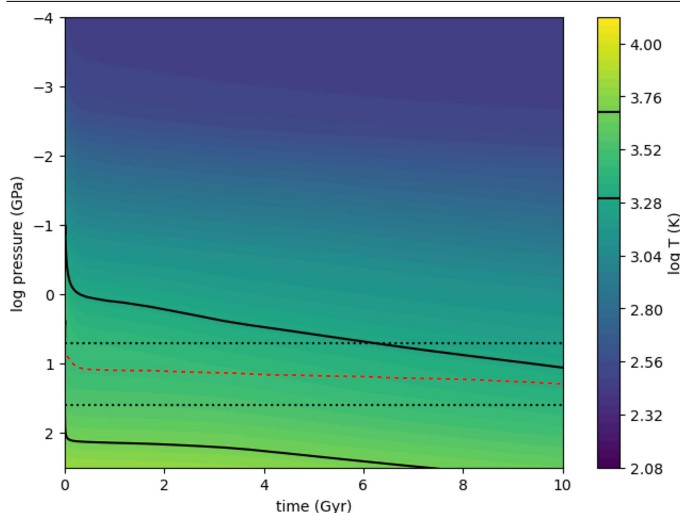

**Extended Data Fig. 2 | Thermal evolution of a rocky 5 $M_E$ sub-Neptune planet with an H + He (5 wt%) envelope.** Shown is the temperature (color) as a function of pressure ($y$-axis) in the interior from the center up to 1 bar pressure, as a function of time ($x$-axis). The range of pressure (dotted black) and temperature (solid black) in which water production is expected according to the experiments is shown. The red dashed line signifies the mantle-envelope interface. Model is based on ref. 28.

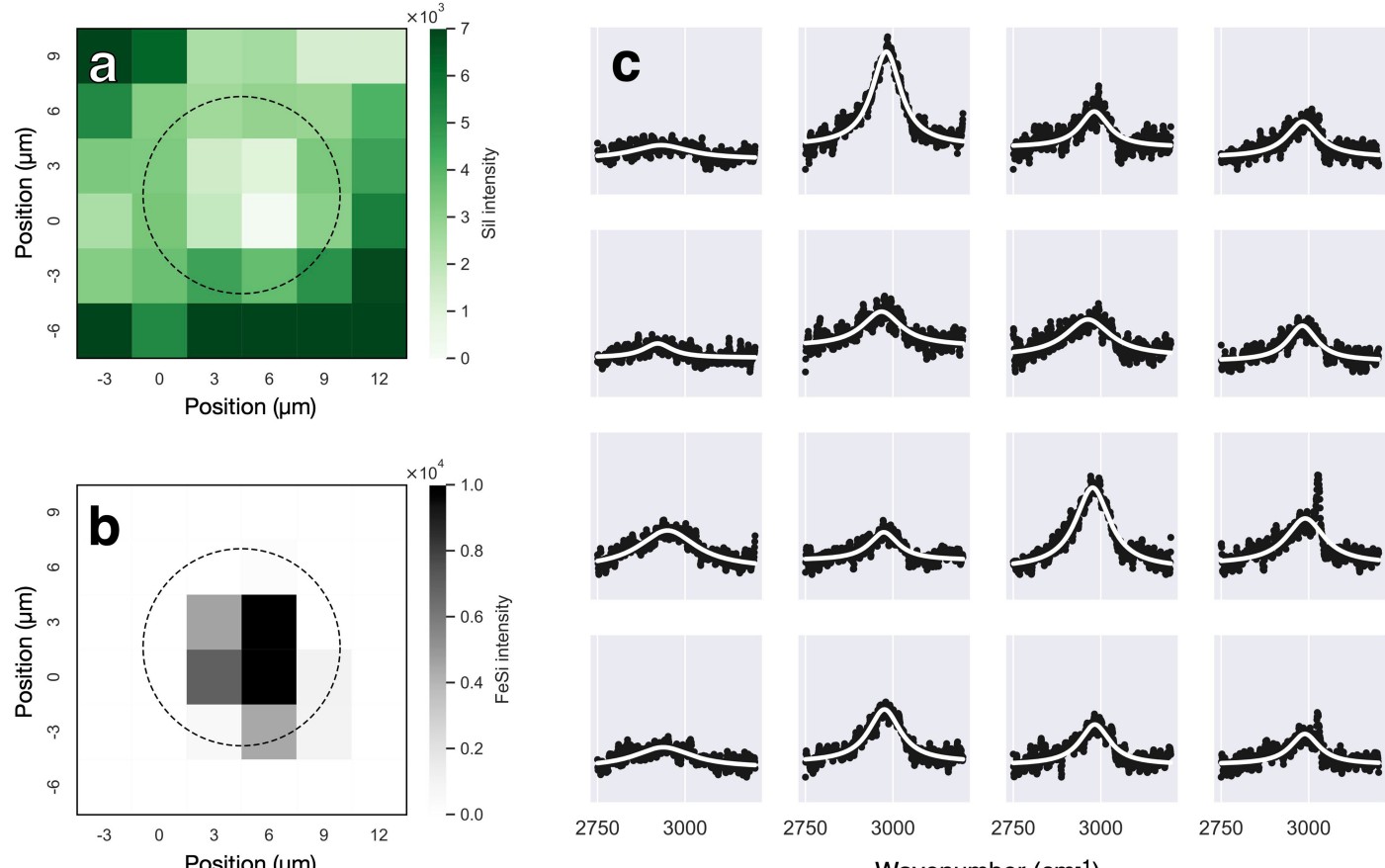

**Extended Data Fig. 3 | Analysis of the samples after heating in a hydrogen medium in LHDAC.** Two dimensional maps of the XRD intensities of **a**, silicates and **b**, B2 $Fe_{1-y}Si_y$ after heating the sample in SCO-11. The anti-correlation between the two at the heating spot center (dashed circle) shows that when melted silicates (bdm) break down, Si is reduced to form $Fe_{1-y}Si_y$. **c**, Raman-active OH vibration from $H_2O$ ice after heating silica + Fe metal. The Raman spectra were measured for a $20 \times 20\,\mu m^2$ heated area. The distance between the spots where the spectra were collected is 5 μm. The spectra were collected after laser heating at 14 GPa.

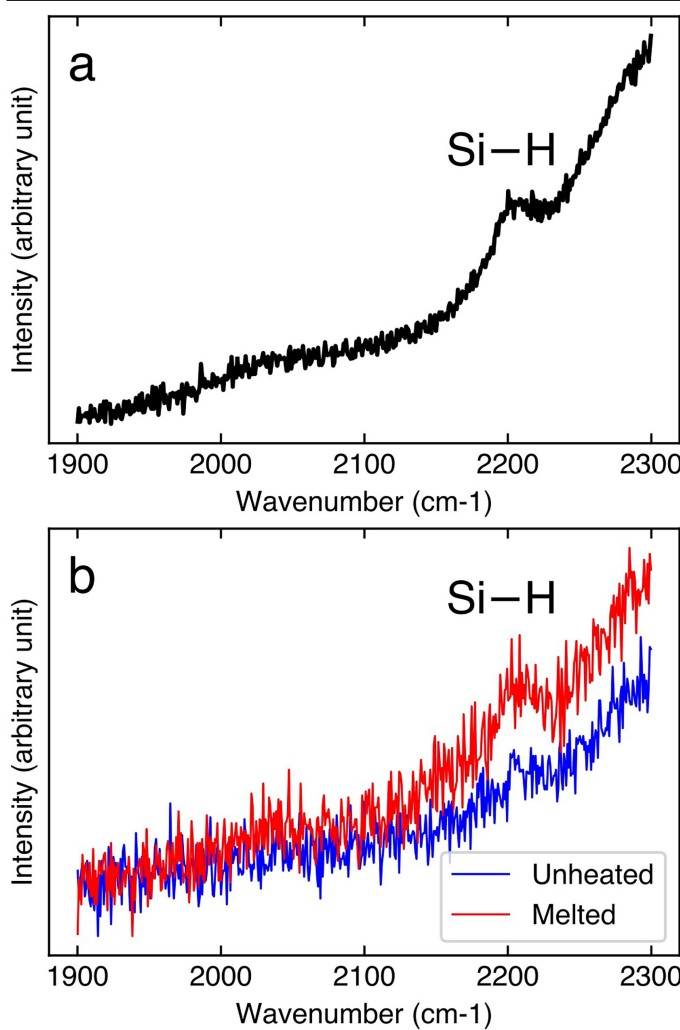

**Extended Data Fig. 4 | Raman spectra from run FAY-8 where starting
fayalite was heated to 1550 K at 14 GPa.** The measurements were conducted
after temperature quench to 300 K and decompression to 2.5 GPa. The Si–H
vibrational mode was detected at the melted area (**a**). No such feature was
observed outside the melted area (blue, **b**). In **b**, we also include spectrum
measured at the melted spot (red) for the same exposure time. The same mode
has been documented for silica melted in hydrogen at 2–3 GPa (ref. 26).

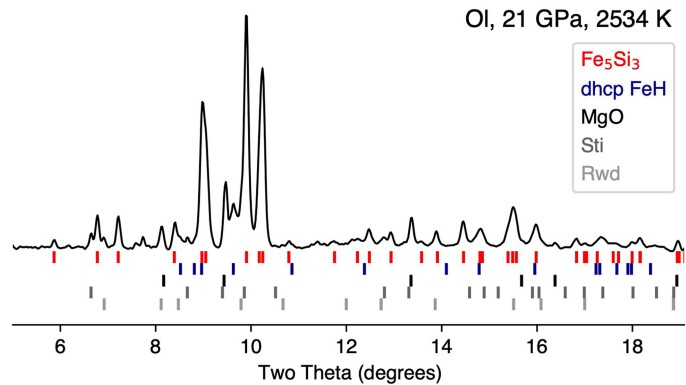

**Extended Data Fig. 5 | X-ray diffraction pattern measured after heating a starting mixture of San Carlos olivine and Fe metal to 2534 K at 21 GPa (run SCO-7).** The ticks below the diffraction pattern are the peak positions of the observed phases. The names of the phases in the legend is ordered same as the ticks from top to bottom. X-ray energy is 37 keV.

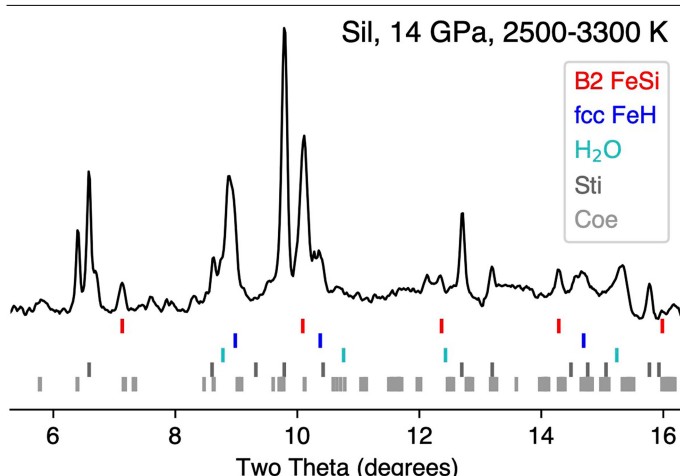

**Extended Data Fig. 6 | X-ray diffraction pattern measured after heating a silica starting material in a hydrogen medium to 2899 K (below the melting temperature) at 14 GPa (run SIL-2).** The ticks below the diffraction pattern are the peak positions of the observed phases. The names of the phases in the legend is ordered same as the ticks from top to bottom. X-ray energy is 37 keV.

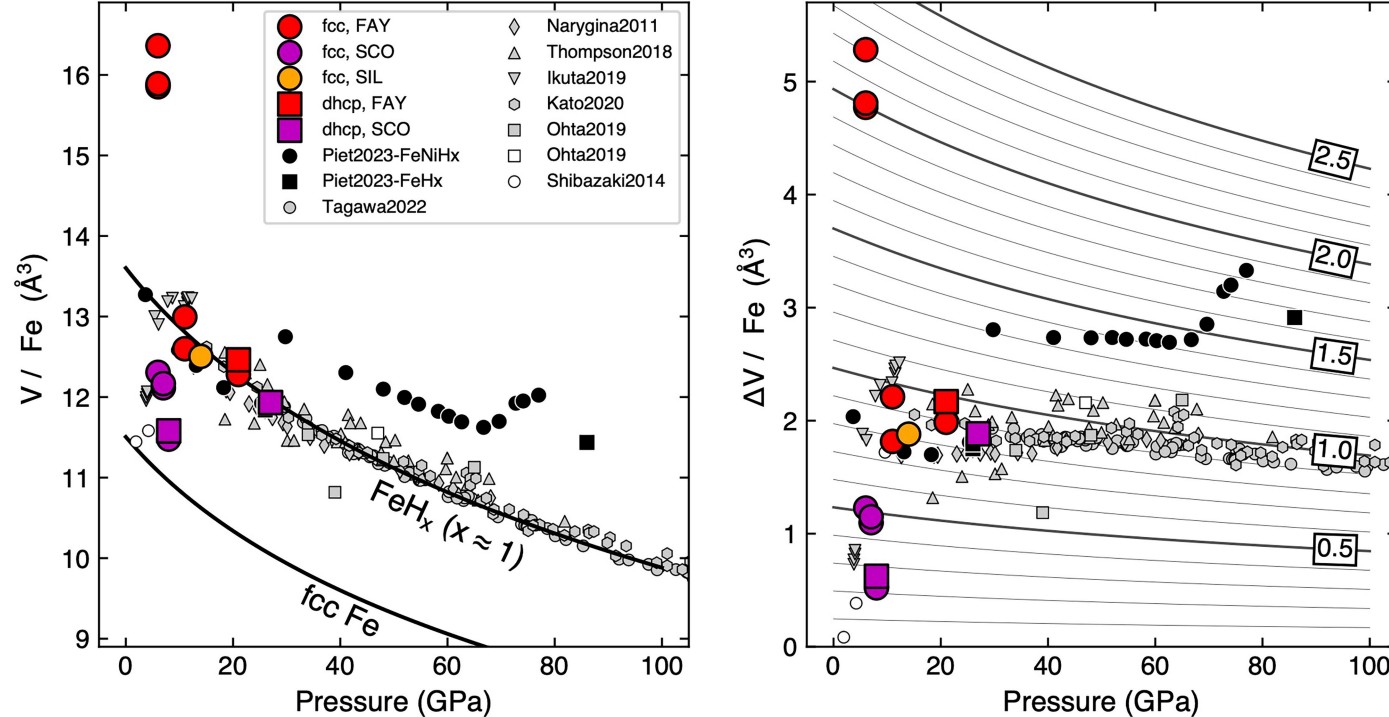

**Extended Data Fig. 7 | Atomic volume (left) and volume increase by H incorporation, $V$ (right), of $FeH_x$ observed in different runs (colored symbols).** The volumes were measured after heating at 300 K. For comparison, the figures also include data points from previous studies[67-73]. The equations of state for fcc Fe (H/Fe = 0) and fcc FeH (H/Fe = 1) are from ref. 74 and ref. 69, respectively. The data points measured for the solid phases quenched from (Fe,Ni)-H liquid are shown as black symbols[75]. The concentration curves ($x$) shown in the right figure are from the density functional theory calculation in ref. 75.

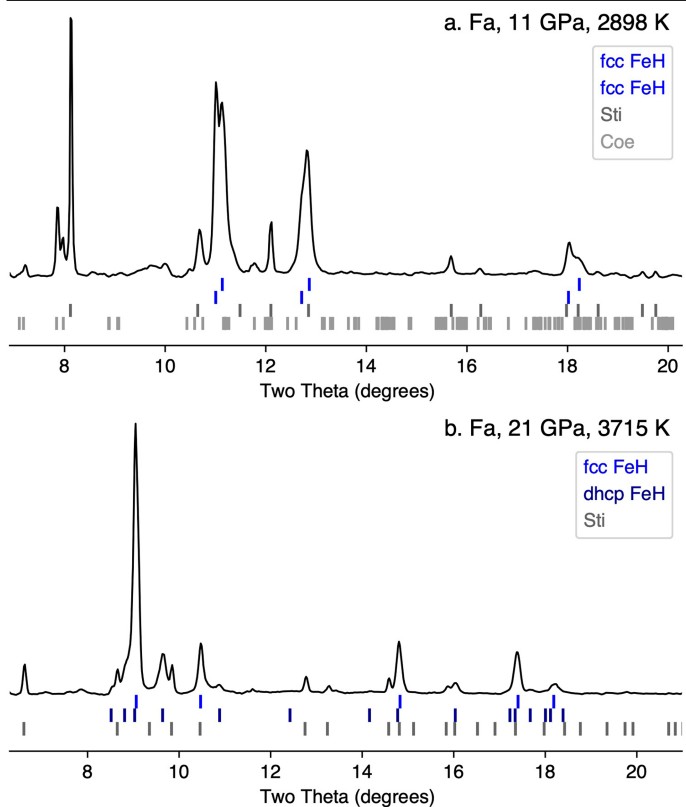

**Extended Data Fig. 8 | X-ray diffraction pattern measured after heating a fayalite starting material in a hydrogen medium to a, 2898 K at 11 GPa (run FAY-3) and b, 3715 K at 21 GPa (run FAY-5). a** was measured at 6 µm away from the heating center. Two separate fcc phases with different volumes (and therefore different levels of hydrogenation) are observed, likely because of different rate of temperature decrease during quenching at different spots and loss of hydrogen during quench of FeH$_x$ melt[75]. The ticks below the diffraction patterns are the peak positions of the observed phases. The names of the phases in the legend is ordered same as the ticks from top to bottom. X-ray energy is **a**, 30 keV and **b**, 37 keV.

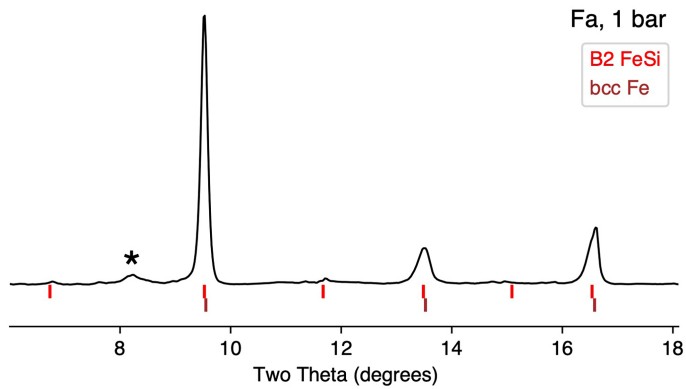

**Extended Data Fig. 9 | X-ray diffraction pattern measured after the decompression of a fayalite starting material heated in a hydrogen medium to 3017 K at 6 GPa (run FAY-1).** The diffraction pattern was measured at 1 bar and 300 K. The ticks below the diffraction pattern are the peak positions of the observed phases. The names of the phases in the legend is ordered same as the ticks from top to bottom. "*" indicates a feature from detector defects. X-ray energy is 37 keV.

**Extended Data Table 1 | Experimental runs in this study**

| Run | $P$ (GPa) | $T$ (K) | A.M. | Result |
|---|---|---|---|---|
| | | $(Mg_{0.9}Fe_{0.1})_2SiO_4 + Fe + H$ | | |
| SCO-1 | 8 | 3042(159) | XRD, SEM | $Fe_{1-y}Si_y$, MgO |
| SCO-2 | 6 | 2833(177) | XRD, SEM | $FeH_x$, MgO |
| SCO-3 | 8 | 3104(269) | XRD, SEM | $Fe_{1-y}Si_y$, $FeH_x$, MgO |
| SCO-4 | 7 | 3050(100) | XRD, SEM | $Fe_{1-y}Si_y$, $FeH_x$, MgO |
| SCO-5 | 7 | 2777(100) | XRD, SEM | $Fe_{1-y}Si_y$, $FeH_x$, MgO |
| SCO-6 | 17 | 3074(267) | XRD | $Fe_5Si_3$, $Fe_{1-y}Si_y$, $FeH_x$, MgO, Rwd, Wds |
| SCO-7 | 21 | 2534(218) | XRD | $Fe_5Si_3$, $FeH_x$, Sti, MgO, Rwd |
| SCO-8 | 21 | 2738(119), 2989(269) | XRD | $Fe_5Si_3$, $FeH_x$, Sti, MgO |
| SCO-9 | 22 | 3101(462) | XRD | $Fe_5Si_3$, $Fe_{1-y}Si_y$, $FeH_x$, Sti, MgO, Rwd |
| SCO-10 | 30 | 3298(150), 2434(100), 2264(100), 2187(100), 2310(100), 2539(100), 4035(144) | XRD, SEM | $Fe_{1-y}Si_y$, $FeH_x$, MgO, Bdm |
| SCO-11 | 37 | 3557(677) | XRD | $Fe_{1-y}Si_y$, $FeH_x$, MgO, Bdm |
| SCO-12 | 42 | 2704(150), 2467(150) | XRD, SEM | $FeH_x$, Bdm, MgO |
| SCO-13a | 42 | 2792(162), 2725(133), 3075(150), 3197(269) | XRD, SEM | $FeH_x$, Bdm, MgO |
| SCO-13b | 42 | 3789(378), 3352(174), 3924(258) | XRD, SEM | $Fe_{1-y}Si_y$, $FeH_x$, MgO |
| | | $Fe_2SiO_4 + H$ | | |
| FAY-1 | 6 | 3017(402) | XRD, Raman, SEM | $Fe_{1-y}Si_y$, $FeH_x$, Fe |
| FAY-2 | 6 | 2254(100) | XRD, Raman | $Fe_{1-y}Si_y$, $FeH_x$, Fe |
| FAY-3 | 11 | 2898(427) | XRD, Raman, SEM | $FeH_x$, Sti, Coe |
| FAY-4 | 11 | 2808(475), 2584(166) | XRD, Raman | $FeH_x$, Sti, Coe |
| FAY-5 | 21 | 3965(407), 3715(954) | XRD, Raman | $FeH_x$, Sti |
| | | $Fe_2SiO_4 + H + Ar$ (50% $H_2$) | | |
| FAY-6 | 10 | 3500(350) | XRD, Raman | $FeH_x$, Sti, Coe |
| FAY-7 | 10 | 1100(500) | XRD, Raman | $FeH_x$, Sti, Coe |
| FAY-8 | 14 | 1550(150) | XRD, Raman | $FeH_x$, Sti, Coe |
| | | $SiO_2 + Fe + H$ | | |
| SIL-1 | 14 | 1934(526) | XRD, Raman | $Fe_{1-y}Si_y$, $FeH_x$, Sti, Coe |
| SIL-2 | 14 | 2899(282) | XRD, Raman | $Fe_{1-y}Si_y$, Sti, Coe |
| SIL-3 | 14 | 2750(100) | XRD, Raman | $Fe_{1-y}Si_y$, $FeH_x$, Sti |
| SIL-4 | 39 | 2149(514) | XRD | $Fe_{1-y}Si_y$, Sti |
| SIL-5 | 39 | 1647(258), 2906(254), 3039(284), 3009(100), 3103(100), 3232(206) | XRD | $Fe_{1-y}Si_y$, Sti |

Estimated uncertainties for pressure ($P$) are 10% (see the Methods section). In some runs, multiples of heating events ($10^5$ laser pulses at 10 kHz) were performed and therefore temperature for each individual heating events are provided. $Fe_{1-y}Si_y$ was observed in the B2 structure throughout these runs. $FeH_x$ appears in the face-centered cubic (fcc) structure during heating and temperature-quenching. When Fe metal was loaded as a starting material (SCO and SIL runs), double hexagonal close packed (dhcp) structured $FeH_x$ was also observed. Fe metal (Fe) was observed as body-centered cubic (bcc) at pressures lower than 10 GPa in some FAY runs. A.M.: analytical methods; XRD: X-ray diffraction; SEM: scanning electron microscopy; Raman: Raman spectroscopy; Rwd: ringwoodite; Wds: wadsleyite; Bdm: bridgmanite; Sti: stishovite; Coe: coesite.