## [Peer Review file · Nature]

Building Wet Planets via High-Pressure Magma-Hydrogen Reactions

Corresponding Author: Dr Harrison Horn

Version 0:

Reviewer comments:

Referee #1

(Remarks to the Author)

This is an interesting paper that presents an important pathway for production of water in planetary systems and for getting it into rocky and icy planets. The authors call on thermodynamics under high temperatures and high pressures to identify conditions within planets where, in the presences of hydrogen, silicates break down to metals and oxygen, and the oxygen combines with hydrogen to form water. This process could put large amounts of water into the “cores” of otherwise dry planets turning them into water worlds. I was unaware of this idea before reading this paper and I find it very interesting and relevant to understanding the origin of Earth’s water, among other things.

The experimental work is elegant and is well described. But the discussion of how these data apply to Earth-like planets, super-Earths, and sub-Neptunes needs some work. Pages 6 through 8 of the paper (the discussion) have not been given the same attention as the Introduction, Methods, and Results. The paper reads like it was written by different authors and they did not get together to reconcile their text and Figures. There is unnecessary repetition, the text and Figures (especially Figure 4) are not always presenting/discussing the same information. This makes the last part of the paper hard to follow. I have no issues with the content of the paper. The authors make a good case that water does not have to condense in cold regions in order to get into the terrestrial planets and ice giants. I had not heard this idea before, and since I am interested in the origin of Earth’s water, I find the idea fascinating. I will be interested to see how it plays out with more-detailed modeling, but these models will just flesh out the details. That basic model seems quite solid.

In the rest of my review, I will mostly be talking about presentation of the ideas. Comments will be tied to page and line numbers.

Once the authors clean up the presentation, this paper will be ready to publish.

Abstract, line 10: Could you please define “mass-radius” for those of us not in your field? I realize it is a standard term (I looked it up on the internet), but please introduce it for the non-expert reader.

Figure 2: This Figure does not always work. Figure 2a is OK, the caption and the legend clearly describe the same phases. Figure 2b, which shows the breakdown products of Fayalite, also has the caption and legend in agreement. Figure 2c: I find this one confusing. Bridgmanite is the high-pressure polymorph of enstatite and would have a composition of MgSiO_3 . Periclase is MgO and ferropericlase is $(\text{Fe},\text{Mg})\text{O}$. The caption says that these two phases appear at 42GPa and below the melting point. The legend shows fcc FeH, MgO (which I assume is the same as ferropericlase but stripped of its Fe, making it periclase (?), and bridgmanite. I assume Bdm is bridgmanite, but that could be made specific in the caption. The figure is supposed to have three colors, but on a printed pdf, I can only see blue (for fcc FeH) and two almost black sets of lines that are indistinguishable in color. I assume that bridgmanite is the bottom of the three sets of lines because it has the more complicated formula and structure. Confirmed in Fig1 2d. Figure 2d supposedly also has bridgmanite and ferropericlase. The patterns are now distinguishable, but the bridgmanite pattern in “d” does not match that in “c”. It does not help that the “Two Theta” scales are different. Maybe this kind of difference between patterns doesn’t matter. But when someone who does not look at X-ray diffraction patterns every day looks at the patterns, it is hard to know what is important and what isn’t. Is there a way to clean this up?

Page 5, lines 116-118: The sentence says that under the same conditions, fayalite results in ten times more Fe being reduced than for San Carlos. The last phrase of the sentence implies that this is because Fe is easier to reduce than Si⁴⁺. This phrase may be out of place. It seems to me that the reason ten times more Fe is reduced when starting from fayalite is that fayalite has almost exactly ten times more Fe in it than does San Carlos olivine. Is that not right? The Mg²⁺ is the hard one to reduce, isn't it? Probably a small change in the text would clear up my confusion.

Page 6 and onward: This is effectively the "Discussion" in the paper. The transition to discussion is smooth, but there are a few details that could have been established earlier in the Introduction or the Results — see below.

Page 6: Definition of "core". It is good that you make this definition explicit, but you have already been using it on previous pages. Shouldn't this definition be put right after the first use, somewhere in the Introduction?

Page 6: This is where you move from describing the experiments to showing how they are relevant to the early solar system. This discussion is somewhat repetitive of material that came earlier. I realize that you are trying to put the previous material into context. But I think the repetition could be cut down. Also, the equations seem out of place. Shouldn't equations 1 and 2 appear earlier? You effectively give their content in the text but the equations themselves in the "results" would seem to be appropriate. Equation 3 is appropriate here because it is the basis for the model that you discuss over the next couple of paragraphs.

Page 6, line 145: Here is where the paper starts to get quantitative about how much hydrogen might be produced in a planet by your mechanism, and puts the model reactions into context. It starts with a composition for a sub-Neptune of 34.9 wt% Fe metal, 62.6 wt% MgSiO₃, and 2.5 wt% H₂. What are these numbers and where did they come from? Do they come from an estimate of cosmic abundances, or estimates of core composition of a planet based on density, moment of inertia, and high-pressure experiments, or ? I realize that at some level it doesn't matter. But a sentence adding this context would make the paper more friendly to readers.

Page 6, lines 145-146: I was able to reproduce the numbers given here, more or less. I got a different number for FeSi (see below), but matched MgO and H₂O within uncertainties of the input data.

Starting composition	Final composition
Fe metal 34.9 wt%	FeSi 34.9 + 0.24211 × 62.6 = 50.06 wt%
MgSiO ₃ 62.6 wt%	MgO 0.40148 × 62.6 = 25.13 wt%
H ₂ 2.5 wt%	H ₂ O = 2.5 / 0.111898 = 22.3 wt%
From paper	FeSi = 52.4, MgO = 25.1, H ₂ O = 22.3

Mg = 24.3050, Si = 28.0855, O = 15.9994, H = 1.00794

$Mg/(Mg+Si+3*O) = 24.3050/(24.3050+28.0855+3*15.9994) = 0.24211$
 $MgO/(Mg+Si+3*O) = (24.3050+15.9994)/(24.3050+28.0855+3*15.9994) = 0.40148$
 $Si/(Mg+Si+3*O) = 28.0855/(24.3050+28.0855+3*15.9994) = 0.279768$
 $H_2/H_2O = (1.00794*2)/(1.00794*2+15.9994) = 0.111898$

Page 6, last full paragraph: This paragraph refers to Fig 4b twice. I think you meant to reference Fig. 4c.

Page 6, line 155-158: I am not following this sentence: The plot in Fig. 4c shows Radius (relative to Earth) plotted against Mass (again relative to Earth). The basic model is shown in Fig. 4c by the two thick lines almost on top of each other at the bottom of the figure. You explain why the lines are on top of each other. You then added 2% H and 20% H₂O to the cores modeled by the thick lines. The results are plotted as the thin red and blue lines. These lines are above the thick lines showing the base model. That would seem to mean that the radius increases with the addition of H or H₂O. The sentence says that if all three involved components are considered (metal layer, silicate layer, H or water – are these the three?), the reaction produced a dramatic decrease in radius for a given mass. But in both cases the composition with H or H₂O show an increase in radius. I do not understand. What am I missing?

Page 7, lines 164-171: I am not following what you are doing in this paragraph. I can reproduce the numbers (29.1 wt% water from 3.3 wt% hydrogen and 16.6 wt% water from 1.9 wt% hydrogen). This is simply the change in wt% that comes from adding O to H₂ (the factor is 8.9). But what does the Mg/Si ratio have to do with this calculation? I cannot connect the Mg/Si = 0.5 and Mg/Si = 2 to either the hydrogen numbers or the water numbers. And these ratios have "more" or "less" Si⁴⁺ than what? Something is missing. It is also confusing to say that Mg/Si is constant and then change it for the calculation. I presume that you mean for a given initial condition, Mg/Si is constant. But none of this is clear.

Page 7, line 191: I think you mean Figure 4c at the beginning of this sentence.

Page 7, line 192: Here you give what you are assuming for the bulk composition of the planet. Great. But this information should also have been provided earlier in the paper.

Page 7, last paragraph: This paragraph seems to have been written with a different figure in mind. The paper has already discussed Fig. 4c (called Fig. 4b in the paper). The text here says that "Fig. 4b (c?) shows the calculated mass-radius relations of H-dominated and H₂O dominated sub-Neptunes with cores with an Earth-like bulk composition along with the observed distribution of the mass and radius of sub-Neptunes up to about half the radii of Uranus and Neptune." Figure 4c shows exoplanets with radii up to slightly larger than Uranus and Neptune, not half the radii.

The paragraph further says, "At 300K, the observed radius variation in sub-Neptunes can be almost completely accounted for between the mass-radius relations of H-dominated and H₂O dominated versions of sub-Neptunes." I am not sure what I should be looking at. The exoplanets plot over much of the diagram from M/M_{Earth} of 4 to 20. There is nothing special about "half the radius of sub-Neptunes" in the plot. The red and blue lines do pass through the field of exoplanets, but the one place the lines seem to have anything to do with the exoplanets is where the thick red and blue curves overlap completely. There seems to be no relationship between the exoplanets and the solid light red and blue lines, except that some planets plot where the lines are. The lines do not cover most of the data, or show potential limits to the range of the data, or anything else. The issue is that the figure does not show what it is described to show, probably because the discussion is referring to a figure that is not in the paper.

The Methods section seems to be comprehensive and well written. But I am not an expert in the techniques being used.

The extended data also seems to be comprehensive. But again, I am not an expert and cannot comment on these discussions and figures in detail.

Referee #2

(Remarks to the Author)

The manuscript provides experimental evidence for the reduction of Si and Fe in silicate melt to metallic phases in hydrogen medium at high pressure and temperature. These are challenging experiments and the experiments have conclusively demonstrated the formation of the metallic phases by in-situ X-ray diffraction measurements. The observed reaction forms the basis for the formation of a H₂O-rich planet from a hydrogen atmosphere. My main concern of the manuscript is that the observations are rather descriptive, and its impact is limited without providing an in-depth thermodynamic model to quantitatively understand the process of the formation of a H₂O-rich planet from a hydrogen atmosphere. It has been demonstrated that hydrogen can be an effective reducing agent for reduction of silicates and oxides; it has also been documented the reaction of Fe and H₂O to form iron hydride at high pressure and temperature. The direction of the reaction strongly depends on the thermodynamic conditions. The new experiments provide some useful constraints on the H-related reactions at high pressure and temperature, but the fundamental understanding of these reactions is incomplete without a comprehensive thermodynamic model to quantitatively describe the systems. These reactions must be quantitatively described in terms of pressure, temperature, composition, and oxygen (or hydrogen) fugacity, which would make the applications more quantitative, particularly defining the boundary conditions for various scenarios. Another important issue which is not adequately discussed in the manuscript is the equilibrium of the reactions with pulsed heating.

Referee #3

(Remarks to the Author)

This paper presents a novel and provocative overarching picture: that reduction reactions between an H₂-rich atmosphere and a molten planet can convert a planet from an H-rich atmosphere to a water-rich planet. Hence, a portion of the mass-radius distributions among exoplanets might be explained through a varying hydrogen/water ratio. So, the general interest of the overarching result is likely high and the mechanism is creative.

That said, I am not clear that the experiments presented, coupled with dynamic considerations, lead robustly and inconclusively to that overarching picture. The experiments themselves are very challenging--pulsed laser heating of H₂-rich samples to minimize hydrogen diffusion into the anvils, and the authors appear to have done a thoughtful job with their analysis. However, the particular experiments involve (1) abundant metallic iron (a proxy core material), (Mg_{0.9}Fe_{0.1})₂SiO₄ olivine (a decent terrestrial mantle proxy), and what appears to be effectively infinite hydrogen; (2) Fe₂SiO₄-olivine (an endmember) + hydrogen; and (3) SiO₂ (a simple model silicate), iron, and hydrogen.

At the range of pressures and temperatures interrogated (~6-40 GPa, ~2000-4000 K), Reaction 1 generally produces reduced iron silicides, iron hydride and MgO; reaction 2 produces iron hydride, and sometimes iron silicides and sometimes SiO₂; and reaction 3 produces iron silicide, sometimes iron hydride, and SiO₂. Once silicon or divalent iron are reduced, oxygen is released, and Raman spectroscopy shows that water is formed in the samples.

Here's where I have one dilemma: although I recognize that the authors' exoplanet definition of "core" on lines 126-127 includes both the rocky/silicate and iron-rich components, the discussion on lines 124-144 elides from discussing hydrogen reaction with a silicate melt (lines 124-130), to apparently requiring reduced iron metal and molten silicates to be juxtaposed to make the reaction run to its fullest potential. That's in accord with their experiment types (1) and (3). So, for this reaction to run the way it's written, it seems that the iron-alloy core (using core in the terrestrial planet sense) has to be either unformed, or in the process of formation, and fully in reactive equilibrium with the silicate of the magma ocean. That's at the crux of the calculation on lines 145-147, where the model sub-Neptune goes from 0-22 wt% water via reaction #3. The issue here is that the iron alloy core appears to achieve equilibrium with the atmosphere of the planet (and the magma ocean, as well)---and that requires some substantially more extensive explanation. For example, perhaps the standard picture of gas-giant planetary evolution involves accretion of an H₂-envelope around a rocky (rock plus iron) core: versions of this are preserved in recent models and interpreted observations (e.g., Alibert et al., *Nature Astronomy*, 2, 873, 2018; Weiss and Bottke, *AGU Advances*, 2, 2021). Given the rapid time-scale of core formation, I'm not sure I see how iron, a magma ocean, and an atmosphere can be fully equilibrated with one another, as Eqn. 3 seems to imply. At a minimum, an extensive discussion of the structural limitations on Eqn. 3 going to completion seems mandated.

My second major issue is also related to the mechanics of how these reactions might proceed, and is encapsulated by the statement on line 127 that “Large amounts of ingassing of hydrogen is possible through physical mixing^{15,16} (sic).” Later in the paper, it is stated (line 211) that the reactions proposed might result in a “fuzzy” boundary between the atmosphere and magma ocean in the “late stage of the conversion” because of H₂O-MgO mixing. But in the early stages, the idea that (say) buoyant MgO might form a proto-crust/septum between the atmosphere and magma ocean, and hence impede reactions that involve atmosphere/magma-ocean equilibration, could also occur. This scenario is pretty much what was published by all of these authors (plus one other) in their Figure 5 in reference 19 (The Planet. Sci. J., 2023), about which they stated “If a FeO-depleted layer forms quickly at the topmost part of the oxide magma ocean where magma reacts with dense hydrogen fluid, it can crystallize and form solid crust, which could shut down reaction 3 and therefore limit the ingassing of H (Figure 5).” Similar caveats, beyond a clause in the abstract (‘in conjunction with internal dynamics’, line 18-19) and a mention that investigations of dynamics are needed (line 188), don’t seem to be present here. My point here would be that there are multiple ways to impede atmosphere-magma ocean equilibration through stratification via the proposed reaction, from formation of an MgO layer, to Fe-depleted magma rising and stratifying, to buoyant H₂-saturated magma stratifying in the near-surface over denser, unsaturated magma. Such effects, which constitute a rather major caveat to the ability of the proposed reactions to dramatically alter a planet’s chemistry, also need to have much more extensive discussion.

My net take on the paper is that the experiments are difficult, timely and provocative. However, the grand scenario proposed in the paper, and presented as a straightforward means of moving from H₂-dominated to H₂O-abundant sub-Neptune planets, has some quite major mechanistic difficulties, which are scarcely, if at all, described. In short, it looks like there’s a bunch of ways that this scenario might well not work, or might work minimally. These include early silicate-iron differentiation (apparently) and interaction-induced stratification at the top of the magma-ocean. That neglect of major caveats doesn’t give the full story to the interested communities. As such, I believe the paper needs to be largely reworked before it can be considered for publication.

Version 1:

Reviewer comments:

Referee #1

(Remarks to the Author)

As I indicated in my initial review, this is an interesting paper that is presenting a new idea about how planets might have gotten their water. The paper is based on some innovative and challenging experimental work that investigates the environment in the interior of sub-Neptunes and super-Earths. They show that it is possible to generate water inside of a nominally anhydrous planets through reduction of SiO₂ to Si₄₊ with the resulting oxygen combining with hydrogen to form water. This idea is new to the planetary science community and, if correct, would significantly change the way we view planet formation and how we think the planets got their water.

The new version of the paper is more focused to address the primary contribution of this work. The text is easier to read and better organized than the original.

The authors took the reviewer’s comments very seriously and provided good responses to each of the comments. The authors addressed each of the comment in detail. Although I am not an expert in this experimental area, I found the discussions with the reviewers to be productive and the authors took those comments seriously. I learned a lot from the responses to the reviewers.

The current paper does not answer all of the questions from the reviewers. But it is not realistic to expect complete answers. This is a new field and this project is at the state of the art. I think the authors have done a good job of explaining the experimental techniques and interpreting their data. They point the way for the next generation of experiments (which will be difficult, but not impossible). The work is mature enough to introduce it to the world and have the community study the new ideas.

My suggestions at this point are basically cosmetic.

For example, I don’t understand the page numbers and section numbers.

Figures 2 and 3 are easier to understand than in the first version.

Figure 4 caption, last sentence – add an “s” to “show”. (diagram shows)

After taking a final look at the entire paper, I think the paper will be ready to publish.

(Remarks on code availability)

Referee #3

(Remarks to the Author)

As with the prior version on the paper, this paper presents some quite provocative observations about reactions between silicates (±iron) and hydrogen at high pressure and temperature conditions, with possible relevance to endogenous water production in hydrogen-mantled planets. The discussion of the data is improved in this version, and (perhaps as an illustration of how difficult to monitor, or likely to evolve in their chemical complexity these reactions are) a new and possibly critically important ingredient/reaction product is introduced in this version of the paper relative to the previous manuscript version: SiH₄. Reactions involving this component do not require equilibration with iron (in my original review, I was skeptical that reactions involving metallic iron would inevitably occur between iron and the overlying H₂ layer, due to

likely rapid core-mantle segregation processes). So, the introduction of this new component/reaction allows water-forming reactions with a silicate layer that need not contain iron metal. That's probably more plausible than the initial scenario.

So, on the positive side, the inclusion of lines 181-201 (and the two paragraphs following) do improve the paper. The accompanying Figure 4 is also a positive addition, but only addresses the possible evolution of an H₂O-rich septum between the silicate magma region and H₂/He atmosphere (which lines 181-183 note would markedly impede the proposed H₂O-forming reactions).

That said, my major concerns with the paper remain dynamical in nature. In my view, the new result --silicon hydride production, and to a lesser extent MgH₂, as described on the top of page 14 of the rebuttal but minimally mentioned (line 52) in the main manuscript, may well produce a supplementary suite of density stratification issues that are not discussed in the current manuscript. Any possible stratification associated with the new reaction products (e.g., a SiH₄/MgH₂-hydride containing H₂O/H₂ layer) is not mentioned in the manuscript. Indeed, while H₂-H₂O miscibility is proposed to enhance mixing between layers (lines 193-197) through buoyancy forces, the negative buoyancy forces and possible stratification that are likely to be associated with possible SiH₄ and MgH₂ mixing/solubility in either the H₂O-rich layer (plus/minus H₂) seem to be undiscussed. I would guess that this could generate more severe density stratifications than are probed in Figure 4, and hence layers that could be stable for longer periods/higher temperatures. So, while I appreciate the authors' much-improved discussion of dynamic effects, I do think that the statement near the base of p. 14 in the rebuttal ("In fact, no clear factors have been identified that would completely shut down endogenic water production") and the calculations of Figure 4 may not be fully reflective of the enhanced chemical complexity that the authors have newly introduced to the water-forming reactions in this version of the paper. My concern here is simple: given the production of Si and Mg-hydrides, Figure 4 may represent a near-best-case scenario for keeping these reactions going in planetary interiors.

My net take is thus that there still need to be some important caveats (and maybe, should the authors choose, additional calculations) introduced in conjunction with statements about the overarching ability of these reactions to, as the title indicates, facilitate "Building a Wet Planet from Dry Materials..." My sense is that the introduction of new reaction products without considering or discussing their effect on the dynamics of the system does motivate revisions to the manuscript that treat their role in generating possible stratification that might occur in evolving planets, and in impeding (or, if there's something creative that I've missed, enhancing) water-generating reactions that might occur between layers/chemical components in their interiors.

(Remarks on code availability)

Version 2:

Reviewer comments:

Referee #3

(Remarks to the Author)

The revised manuscript is substantially improved, and now contains (as I believe is appropriate) more caveats on the scope of the proposed H₂/silicate reactions. As such, I can support its publication in Nature.

The one aspect that might be worth having a comment on, and that I remain not fully compelled by, is what the authors describe in their response as a "relatively conservative scenario," involving what sounds like efficient convective mixing of their reactants (principally H₂) into underlying molten silicates (loosely, lines 177-179, 187-189 and 193-194, and Section 12 of the Extended Data).

In their discussion, they focus on H₂-H₂O miscibility (e.g., line 174), and on likely SiH₄-H₂ miscibility (line 187 and Section 12). I think I'd take a step back here and note that, while there is almost certainly a single fluid at these P/T conditions, it's a single fluid that has three components of different densities and probably different diffusivities. That seems like a recipe for possibly producing double diffusive boundary layers or (more optimistically for the authors' scenarios) oscillatory double diffusive convection. Either would likely impede mixing in a manner that would slow the convective/mixing "efficiency" invoked on lines 167, 186 and 193.

That said, I don't expect the authors to provide an analysis of such effects---but it might well be worth noting that they might exist.

(Remarks on code availability)

Response to Reviewers

L1 Major Changes

L1.1 New Data and Modeling

In this revision, we have included the following new experimental data and modeling results, which further strengthen our main conclusion.

- 1. Modeling of the thermodynamic conditions at the core-envelope boundary (CEB) of sub-Neptunes:** We calculated interior thermal evolution of sub-Neptunes and derived the pressure-temperature (P - T) conditions at the CEB. We demonstrated that our experimental conditions align well with those expected in this region (lines 136–142; Extended Data Text 8; Extended Data Figs 1 and 2).
- 2. Modeling of mixing between hydrogen and endogenic water:** The efficiency of mixing within the envelope can be important for the extent of endogenic water production in sub-Neptunes. Increased water concentration resulting from the hydrogen-magma reaction can suppress or even reverse this reaction. The dynamics model introduced in this revision shows that fluid flow under the high-temperature conditions of the CEB can transport water from the CEB to shallower depths within the envelope, reducing water activity at the reaction zone. This mixing mechanism allows endogenic water production to continue until the system reaches temperatures of ~ 3500 K (lines 181–196; Fig. 4; and Extended Data Fig. 2).
- 3. Experimental evidence for endogenic water production through SiH_4 formation:** In our original submission, we reported water production through Si^{4+} reduction (i.e., Fe-Si alloy formation). During this revision, we identified an additional process for endogenic water production— SiH_4 formation (lines 92–95; Equation 2; Extended Data Text 7; Extended Data Tab. 1; Extended Data Fig. 3). The significance of this finding lies in the following: while water production via Si^{4+} reduction is limited to reducing conditions, SiH_4 formation can occur under much more oxidizing conditions. In planetary contexts, the SiH_4 formation reaction can significantly extend the timeframe for endogenic water production (lines 171–180).
- 4. New experimental dataset acquired under low hydrogen concentration:** In the original submission, we reported experiments conducted in a pure H_2 medium. To explore the reaction under low hydrogen concentrations, we added a new dataset collected in an Ar + H_2 mixture medium, with H_2 concentration at 50%. We found that under hydrogen-limited (or more oxidizing) conditions, SiH_4 formation becomes the primary process for water production (lines 129–133; Extended Data Text 7; Extended Data Tab. 1; Extended Data Fig. 1).
- 5. Semi-quantitative analysis of our experiments:** We have estimated the quantities of reactants and reaction products, including the amount of hydrogen consumed and water produced, in our experiments involving olivine and Fe metal in a hydrogen medium. Although uncertainties exist (Extended Data Text 9), rendering the estimates semi-quantitative, these calculations demonstrate the relevance of our experiments to sub-Neptunes (lines 147–160). A Jupyter notebook for these calculations is provided (Supplementary Code 1).

L1.2 Revised Main Conclusions

We revised the manuscript to reflect the following main conclusion: **A significant amount of water (tens of weight percent) can be produced endogenously through reactions between hydrogen and magma in sub-Neptunes.** Both the abstract and discussion sections have been completely rewritten to support this conclusion.

Key supporting evidence: The main conclusion above is supported by the following lines of evidence from our work:

- Complete removal of Si and substantial release of oxygen from silicate melt due to the hydrogen-magma reaction at hydrogen fugacities consistent with sub-Neptunes (Figs 1 and 2; Supplementary Code 1).
- Sustained water production through SiH_4 formation under oxidizing or low hydrogen conditions.
- Mixing within the envelope that reduces water activity at the reaction zone, thereby extending endogenic water production.

Key implications:

- Water can be produced internally within planets through hydrogen-silicate reactions. Therefore, the detection of water in the atmospheres of close-in transiting exoplanets cannot be solely attributed to large-scale planetary migration, which assumes water is primarily accreted beyond the snow line.
- Hydrogen-world and water-world sub-Neptunes need not form through entirely distinct processes but may be interconnected through the chemical evolution of planets.
- Larger rocky bodies, which accrete significantly more hydrogen during formation, may generate far more water via hydrogen-magma reactions under high-pressure conditions than smaller bodies such as Earth.

L2 Referee #1

This is an interesting paper that presents an important pathway for production of water in planetary systems and for getting it into rocky and icy planets. The authors call on thermodynamics under high temperatures and high pressures to identify conditions within planets where, in the presences of hydrogen, silicates break down to metals and oxygen, and the oxygen combines with hydrogen to form water. This process could put large amounts of water into the “cores” of otherwise dry planets turning them into water worlds. I was unaware of this idea before reading this paper and I find it very interesting and relevant to understanding the origin of Earth’s water, among other things.

The experimental work is elegant and is well described. But the discussion of how these data apply to Earth-like planets, super-Earths, and sub-Neptunes needs some work. Pages 6 through 8 of the paper (the discussion) have not been given the same attention as the Introduction, Methods, and Results. The paper reads like it was written by different authors and they did not get together to reconcile their text and Figures. There is unnecessary repetition, the text and Figures (especially Figure 4) are not always presenting/discussing the same information. This makes the last part of the paper hard to follow.

I have no issues with the content of the paper. The authors make a good case that water does not have to condense in cold regions in order to get into the terrestrial planets and ice giants. I had not heard this idea before, and since I am interested in

the origin of Earth's water, I find the idea fascinating. I will be interested to see how it plays out with more-detailed modeling, but these models will just flesh out the details. That basic model seems quite solid.

We sincerely thank the reviewer for their kind words regarding this experimental work and their candid assessment of the discussion on the applications of our results. As the reviewer noted, these experiments were extremely challenging due to containment issues associated with dense hydrogen fluid. Nonetheless, we are highly encouraged by the successful combination of key technical advancements that allowed us to overcome these difficulties and obtain critical data. As the reviewer highlighted, this work makes a first-order contribution to understanding the origin of water: “The authors make a good case that water does not have to condense in cold regions in order to get into the terrestrial planets and ice giants.” While further details will undoubtedly be of interest, the rapid advancements in detecting exoplanet atmospheres make the “basic model,” which the reviewer described as “solid,” a crucial factor in ongoing efforts to understand the exoplanet survey observations. In light of this feedback, we have decided to focus on the broader implications of our findings (see section L1.2) in this revision. Consequently, we have removed the discussions related to mass-radius relationships to ensure a more streamlined and impactful presentation.

In the rest of my review, I will mostly be talking about presentation of the ideas. Comments will be tied to page and line numbers. Once the authors clean up the presentation, this paper will be ready to publish.

We have addressed each of the line items below. We hope this paper is “ready to publish” as the reviewer evaluated.

Abstract, line 10: Could you please define “mass-radius” for those of us not in your field? I realize it is a standard term (I looked it up on the internet), but please introduce it for the non-expert reader.

The sentence is revised and we provide additional explanation there.

Figure 2: This Figure does not always work. Figure 2a is OK, the caption and the legend clearly describe the same phases. Figure 2b, which shows the breakdown products of Fayalite, also has the caption and legend in agreement. Figure 2c: I find this one confusing. Bridgmanite is the high-pressure polymorph of enstatite and would have a composition of $MgSiO_3$. Periclase is MgO and ferropericlase is $(Fe,Mg)O$. The caption says that these two phases appear at 42 GPa and below the melting point. The legend shows fcc FeH, MgO (which I assume is the same as ferropericlase but stripped of its Fe, making it periclase (?), and bridgmanite. I assume Bdm is bridgmanite, but that could be made specific in the caption. The figure is supposed to have three colors, but on a printed pdf, I can only see blue (for fcc FeH) and two almost black sets of lines that are indistinguishable in color. I assume that bridgmanite is the bottom of the three sets of lines because it has the more complicated formula and structure. Confirmed in Figl 2d. Figure 2d supposedly also has bridgmanite and ferropericlase. The patterns are now distinguishable, but the bridgmanite pattern in “d” does not match that in “c”. It does not help that the “Two Theta” scales are different. Maybe this kind of difference between patterns doesn't matter. But when someone who does not look at X-ray diffraction patterns every day looks at the patterns, it is hard to know what is important and what isn't. Is there a way to clean this up?

As suggested, the revised caption now includes explanations for the abbreviations used in the legend and labels, such as bridgmanite (bdm) and ferropericlase (fp). We have also ensured that the gray color representing bdm matches between panels (c) and (d) in Fig. 2.

Originally, we labeled MgO instead of ferropericlase because all Fe is reduced, effectively transforming ferropericlase, $(Mg,Fe)O$, into periclase, which is MgO . However, we understand the reviewer's suggestion to aim for greater consistency in the naming of the observed phases.

The relative positions of bdm peaks in panels (c) and (d) differ slightly due to variations in Fe content between the bdm phases in these panels.

Regarding the two-theta diffraction angle range, it is challenging to maintain the exact same range across the entire dataset because X-ray energy and the angular aperture of the diamond anvil cell (DAC) vary between experimental runs.

Page 5, lines 116-118: The sentence says that under the same conditions, fayalite results in ten times more Fe being reduced than for San Carlos. The last phrase of the sentence implies that this is because Fe is easier to reduce than Si⁴⁺. This phrase may be out of place. It seems to me that the reason ten times more Fe is reduced when starting from fayalite is that fayalite has almost exactly ten times more Fe in it than does San Carlos olivine. Is that not right? The Mg²⁺ is the hard one to reduce, isn't it? Probably a small change in the text would clear up my confusion.

This sentence (now at line 126) has been revised for clarity.

Page 6 and onward: This is effectively the “Discussion” in the paper. The transition to discussion is smooth, but there are a few details that could have been established earlier in the Introduction or the Results — see below.

Page 6: Definition of “core”. It is good that you make this definition explicit, but you have already been using it on previous pages. Shouldn't this definition be put right after the first use, somewhere in the Introduction?

The definition has been moved to the first usage of the term “core” in line 33.

Page 6: This is where you move from describing the experiments to showing how they are relevant to the early solar system. This discussion is somewhat repetitive of material that came earlier. I realize that you are trying to put the previous material into context. But I think the repetition could be cut down.

The new journal format recently implemented by *Nature* includes section titles. In accordance with this format, we have added section titles to ensure smoother transitions.

Also, the equations seem out of place. Shouldn't equations 1 and 2 appear earlier? You effectively give their content in the text but the equations themselves in the “results” would seem to be appropriate. Equation 3 is appropriate here because it is the basis for the model that you discuss over the next couple of paragraphs.

Since we identified an additional process for endogenic water production, a new chemical reaction has been added (equation 2), resulting in the renumbering of the equations.

Page 6, line 145: Here is where the paper starts to get quantitative about how much hydrogen might be produced in a planet by your mechanism, and puts the model reactions into context. It starts with a composition for a sub-Neptune of 34.9 wt% Fe metal, 62.6 wt% MgSiO₃, and 2.5 wt% H₂. What are these numbers and where did they come from? Do they come from an estimate of cosmic abundances, or estimates of core composition of a planet based on density, moment of inertia, and high-pressure experiments, or ? I realize that at some level it doesn't matter. But a sentence adding this context would make the paper more friendly to readers.

The specific section mentioned by the reviewer has been replaced in this revision with a quantification of reactants and reaction products (see #5 in section L1.1). This calculation is included in the newly added Jupyter notebook (Supplementary Code 1).

Page 6, lines 145-146: I was able to reproduce the numbers given here, more or less. I got a different number for FeSi (see below), but matched MgO and H₂O within uncertainties of the input data.

Starting composition Final composition

Fe metal 34.9 wt% FeSi $34.9 + 0.24211 \times 62.6 = 50.06$ wt%

MgSiO₃ 62.6 wt% MgO $0.40148 \times 62.6 = 25.13$ wt%

H₂ 2.5 wt% H₂O $2.5 / 0.111898 = 22.3$ wt%

From paper FeSi = 52.4, MgO = 25.1, H₂O = 22.3

Mg = 24.3050, Si = 28.0855, O = 15.9994, H = 1.00794

Mg/(Mg+Si+3O) = $24.3050 / (24.3050 + 28.0855 + 3 \times 15.9994) = 0.24211$

MgO/(Mg+Si+3O) = $(24.3050 + 15.9994) / (24.3050 + 28.0855 + 3 \times 15.9994) = 0.40148$

$$\text{Si}/(\text{Mg}+\text{Si}+3\text{O}) = 28.0855/(24.3050 + 28.0855 + 3 \times 15.9994) = 0.279768$$
$$\text{H}_2/\text{H}_2\text{O} = (1.00794 \times 2)/(1.00794 \times 2 + 15.9994) = 0.111898$$

The reviewer's calculation does not yield a total of 100% for the product side; instead, it sums to 97.5%. In contrast, the numbers presented in the paper add up to 100%. To ensure consistency, the number of elements (in moles) must be conserved for the reaction. Subsequently, the weight percentages of the components on both the product and reactant sides should be normalized to achieve a total of 100% for each side. Now we provide all the major calculations in this paper in Jupyter notebooks (Supplementary Codes 1, 2, and 3).

Page 6, last full paragraph: This paragraph refers to Fig 4b twice. I think you meant to reference Fig. 4c.

In this revision, we have chosen to focus on explaining the formation of water-rich sub-Neptunes within the snow line. As part of this new focus, we have replaced the mass-radius diagrams in Fig. 4 with the results of dynamics calculations for mixing (#2 in section L1.1).

Page 6, line 155-158: I am not following this sentence: The plot in Fig. 4c shows Radius (relative to Earth) plotted against Mass (again relative to Earth). The basic model is shown in Fig. 4c by the two thick lines almost on top of each other at the bottom of the figure. You explain why the lines are on top of each other. You then added 2% H and 20% H2O to the cores modeled by the thick lines. The results are plotted as the thin red and blue lines. These lines are above the thick lines showing the base model. That would seem to mean that the radius increases with the addition of H or H2O. The sentence says that if all three involved components are considered (metal layer, silicate layer, H or water – are these the three?), the reaction produced a dramatic decrease in radius for a given mass. But in both cases the composition with H or H2O show an increase in radius. I do not understand. What am I missing?

The figure is now replaced by dynamic simulation results.

Page 7, lines 164-171: I am not following what you are doing in this paragraph. I can reproduce the numbers (29.1 wt% water from 3.3 wt% hydrogen and 16.6 wt% water from 1.9 wt% hydrogen). This is simply the change in wt% that comes from adding O to H2 (the factor is 8.9). But what does the Mg/Si ratio have to do with this calculation? I cannot connect the Mg/Si = 0.5 and Mg/Si = 2 to either the hydrogen numbers or the water numbers. And these ratios have “more” or “less” Si4+ than what? Something is missing. It is also confusing to say that Mg/Si is constant and then change it for the calculation. I presume that you mean for a given initial condition, Mg/Si is constant. But none of this is clear.

The figure is now replaced by dynamic simulation results.

Page 7, line 191: I think you mean Figure 4c at the beginning of this sentence.

The figure is now replaced by dynamic simulation results.

Page 7, line 192: Here you give what you are assuming for the bulk composition of the planet. Great. But this information should also have been provided earlier in the paper.

This section has been removed to align with our revised focus on the significance of forming water-rich sub-Neptunes within the snow line (see section L1.2).

Page 7, last paragraph: This paragraph seems to have been written with a different figure in mind. The paper has already discussed Fig. 4c (called Fig. 4b in the paper). The text here says that “Fig. 4b (c?) shows the calculated mass-radius relations of H-dominant and H2O dominated sub-Neptunes with cores with an Earth-like bulk composition along with the observed distribution of the mass and radius of sub-Neptunes up to about half the radii of Uranus and Neptune.” Figure 4c shows exoplanets with radii up to slightly larger than Uranus and Neptune, not half the radii.

The figure is now replaced by dynamic simulation results.

The paragraph further says, “At 300K, the observed radius variation in sub-Neptunes can be almost completely accounted for between the mass-radius relations of H-dominated and H2O dominated versions of sub-Neptunes.” I am not sure what I should

be looking at. The exoplanets plot over much of the diagram from M/M_{Earth} of 4 to 20. There is nothing special about “half the radius of sub-Neptunes” in the plot. The red and blue lines do pass through the field of exoplanets, but the one place the lines seem to have anything to do with the exoplanets is where the thick red and blue curves overlap completely. There seems to be no relationship between the exoplanets and the solid light red and blue lines, except that some planets plot where the lines are. The lines do not cover most of the data, or show potential limits to the range of the data, or anything else. The issue is that the figure does not show what it is described to show, probably because the discussion is referring to a figure that is not in the paper.

The figure has been replaced with results from dynamic simulations. Furthermore, with the main focus shifting to the formation of water-rich sub-Neptunes within the snow line, we no longer discuss mass-radius implications in the revised manuscript (see section L1.2).

The Methods section seems to be comprehensive and well written. But I am not an expert in the techniques being used.

The extended data also seems to be comprehensive. But again, I am not an expert and cannot comment on these discussions and figures in detail.

Thank you.

L3 Referee #2

The manuscript provides experimental evidence for the reduction of Si and Fe in silicate melt to metallic phases in hydrogen medium at high pressure and temperature. These are challenging experiments and the experiments have conclusively demonstrated the formation of the metallic phases by in-situ X-ray diffraction measurements. The observed reaction forms the basis for the formation of a H₂O-rich planet from a hydrogen atmosphere.

My main concern of the manuscript is that the observations are rather descriptive, and its impact is limited without providing an in-depth thermodynamic model to quantitatively understand the process of the formation of a H₂O-rich planet from a hydrogen atmosphere.

We appreciate the reviewer recognizing the challenging aspects of our experiments and the importance of their planetary implications, particularly regarding the origin of water-rich planets. As the reviewer pointed out, our experiments provide the first observation of an important reaction that “forms the basis for the formation of an H₂O-rich planet from a hydrogen atmosphere,” which we believe represents a paradigm-shifting discovery. This is especially significant given that models and theories have historically assumed water is added primarily through delivery from the outer regions of planetary systems. Our study introduces a new mechanism for producing large amounts of water (tens of weight percent) through chemical reactions between hydrogen and magma, without requiring significant water delivery or planetary migration.

The reviewer raised questions about an “in-depth thermodynamic model” but did not provide specific details. Based on the reviewer’s remaining comments, we infer that they are referring to an equilibrium constant for the reaction determined over a wide range of pressures, temperatures, and redox conditions. Developing such a thermodynamic model requires an analytical technique capable of quantifying hydrogen and water in high-pressure samples. Given the volatility of water and hydrogen during sample recovery, this technique would need to work directly on samples within high-pressure cells. To our knowledge, no such method is currently available.

The key point is that we provide sufficient evidence to support the formation of water-rich planets through chemical reactions between hydrogen and magma, which marks a paradigm shift in our understanding of planetary formation and exoplanet atmosphere measurements. While an “in-depth thermodynamic model” would be beneficial, it would not alter this primary conclusion. We agree that such a model

would be useful for modeling the evolutionary pathways of individual sub-Neptunes, but that is not the primary intent of this paper. In this revision, we have removed the discussion on mass-radius diversity among sub-Neptunes, originally included in our submission, to focus solely on the most directly related implications. This adjustment avoids the need for an “in-depth thermodynamic model,” which is currently not feasible and would require significant experimental advancements in the coming years.

In this revision, we have added new data and models (section L1.1) that further strengthen our main conclusion. Briefly, new calculations (see #5 in section L1.1) demonstrate that hydrogen-magma reactions can produce tens of weight percent water, sufficient to form water-rich sub-Neptunes. Additionally, the newly included planetary structure model (see #1 in section L1.1) shows that the P - T conditions of our experiments overlap with those expected at the CEB of sub-Neptunes. We also note that “the process of the formation of an H_2O -rich planet from a hydrogen atmosphere,” as highlighted by the reviewer, involves not only thermodynamics but also material mixing (i.e., dynamics). To address this, we have added dynamic models (see #2 in section L1.1).

In this revision, we have further advanced our understanding of “the process of formation” by identifying an additional mechanism for endogenic water production, specifically the formation of SiH_4 (Equation 2). In Supplementary Code 1, we provide estimates of reactants and reaction products, along with an uncertainty assessment, addressing the need to “quantitatively understand ...” (see #5 in section L1.1).

We acknowledged and discussed that sub-Neptunes may have water contents lower than the tens of weight percent produced by the endogenic process. This is based on both dynamic factors (see #2 in section L1.1) and thermodynamic conditions (Supplementary Code 1), as discussed in the main text (lines 211–236). Even in cases where water production is suppressed (e.g., rapid cooling of some sub-Neptunes) or water is lost (e.g., strong stellar radiation), endogenic water production under higher pressures at the CEB makes larger rocky planets much more likely to retain significant water compared to smaller rocky planets, such as Earth (lines 226–236 and Supplementary Code 2). This finding has significant implications for the habitability of exoplanets.

It has been demonstrated that hydrogen can be an effective reducing agent for reduction of silicates and oxides ...

Hydrogen’s potential as an effective reducing agent has been considered in previous modeling efforts^{4,5,7,9,22} within the context of planetary science. However, these studies lacked experimental evidence at the P - T conditions relevant to the core-envelope boundary (CEB) of sub-Neptunes (as acknowledged in lines 46–48). At the CEB of sub-Neptunes, hydrogen and water are expected to exist as liquid or supercritical fluid, based on their phase diagrams^{88,97,98}. We emphasize that existing models rely on thermodynamic parameters measured for the gaseous state of hydrogen and water under very low P - T conditions and extrapolation based on “ideal gas” assumption. To our knowledge, our experiment is the first to directly investigate hydrogen-magma reactions under the physical states (liquid state of hydrogen and water) and P - T conditions relevant to the CEB of sub-Neptunes (lines 136–142).

In this revision, we identified and reported an additional reaction contributing to endogenic water production: the formation of SiH_4 . This reaction occurs not because “hydrogen can be an effective reducing agent.” Instead, it arises from Si forming a direct chemical bond with H without altering its oxidation state. The formation of SiH_4 has been documented in a few experimental studies^{25,26,99}, but those experiments were conducted at P - T conditions too low to maintain silicates and metals in a liquid state. These states are critical for understanding the CEB of sub-Neptunes, where all involved materials exist in the liquid phase.

In this revision, we demonstrated that under realistic physical states of the involved materials (hydrogen, water, silicate, and metal) at the P - T conditions expected for the CEB of sub-Neptunes, significantly more water can be produced than previously predicted by models based on extrapolations from unrealistic states (lines 161–170).

... it has also been documented the reaction of Fe and H₂O to form iron hydride at high pressure and temperature.

The reaction between Fe and H₂O to form iron hydride, as mentioned by the reviewer, is not relevant to the focus of this study or the design of our experiments. The experiments cited by the reviewer were conducted under H₂O-saturated conditions (e.g., ref.¹⁰⁰), which are unrelated to the hydrogen-saturated conditions characteristic of hydrogen-rich sub-Neptunes, as discussed in this study.

The direction of the reaction strongly depends on the thermodynamic conditions. The new experiments provide some useful constraints on the H-related reactions at high pressure and temperature, but the fundamental understanding of these reactions is incomplete without a comprehensive thermodynamic model to quantitatively describe the systems. These reactions must be quantitatively described in terms of pressure, temperature, composition, and oxygen (or hydrogen) fugacity, which would make the applications more quantitative, particularly defining the boundary conditions for various scenarios.

In this revision, we demonstrated that our experimental P - T conditions align with those of the target planet group, i.e., sub-Neptunes (#1 in section L1.1), and that the estimated quantities of reactants and reaction products (#5 in section L1.1) are relevant to sub-Neptunes (lines 147–160). Under these conditions, our experiments showed that all Si in olivine was released to form Fe-Si alloy (Equation 1) and SiH₄ (Equation 2), as convincingly illustrated in Fig. 2. These data demonstrate that the “direction of the reaction” strongly favors the reactant side in both Equations 1 and 2 under conditions expected at the CEB of sub-Neptunes, directly addressing the reviewer’s question.

To understand the evolutionary pathways of specific sub-Neptune planets, we agree with the reviewer that a comprehensive description of the hydrogen-silicate reaction over a wide parameter space would be desirable. However, we argue that it is urgent to make the exoplanet and planetary science communities aware that tens of weight percent of water can be generated endogenously, i.e., through reaction between dense liquid hydrogen and silicate magma, unlike far smaller amount of water (2–3000 times) recent modeling studies^{4,5} have suggested for endogenic processes (lines 161–170). The primary limitation of these recent modeling studies is that they rely on long extrapolation of the assumed ideal gas behaviors of hydrogen and water. We emphasize that, at the P - T conditions of the CEB of sub-Neptunes, hydrogen and water exist in the liquid or supercritical fluid state. Our contribution is particularly important because the detection of water in the atmospheres of close-in, low-density exoplanets has been used to infer large-scale planetary migration—a notion directly challenged by our work.

Moreover, hydrogen-rich and water-rich sub-Neptunes have traditionally been considered to form through entirely distinct processes^{41,42}. For the first time (to our knowledge), we identify a clear evolutionary link between these two classes of planets through chemical reactions. Therefore, our key discoveries—the identification of endogenic water production and the significant amounts of water generated through this process—have immediate implications for a range of issues, including the interpretation of exoplanet atmospheric spectra, modeling mass-radius relationships and internal structures of sub-Neptunes, and advancing planet formation theories. Considering the reviewer’s comments, we have decided to focus on the planet-scale endogenic production of water in this paper, rather than the diversity in mass-radius relationships among sub-Neptunes discussed in our original manuscript.

It is also critical to note that the amount of water produced endogenously should be substantially higher in sub-Neptunes and super-Earths, given the sufficiently high P - T conditions for high-density liquid states

of the involved materials. For example, a recent modeling study published in *Nature* (ref.⁹) considered endogenic water production for Earth and estimated only ~ 200 ppm of water. This is because the reaction is far less efficient for gaseous hydrogen and water under the lower P - T conditions expected for smaller rocky bodies.

Of course, we recognize that several factors can influence endogenic water production, which we discussed in this revision (lines 211–236). Even with a complete thermodynamic model, additional factors would need to be considered for individual sub-Neptunes, such as orbital distances, thermal history, mixing, and dynamics. Therefore, in this paper, we focus on the most important and robust conclusion: endogenic water production can yield significant amounts of water, sufficient to convert hydrogen-rich planets into water-rich ones through chemical reactions, regardless of their orbital distances.

Our estimation shows that even if only the uppermost 5% of the core undergoes chemical reactions, endogenic water production would still generate 2–4 wt% H_2O —substantially more than the total water content of Earth (lines 226–236; Supplementary Code 2). The most robust conclusion from these considerations is that water should be far more common among super-Earths and sub-Neptunes than among smaller rocky bodies, such as Earth and Venus.

Another important issue which is not adequately discussed in the manuscript is the equilibrium of the reactions with pulsed heating.

It is unclear which specific aspects of pulsed heating the reviewer referred to. We therefore assume that the reviewer is concerned about the short heating duration. To the best of our knowledge, short-duration heating is the only viable method to achieve the sufficiently high temperatures necessary to melt silicates in a dense liquid hydrogen environment. This is because liquid hydrogen is highly mobile and cannot be effectively contained in a high-pressure apparatus during prolonged heating. We believe that the reviewer's concern may relate to whether pulsed heating could result in the formation of metastable phases or incomplete reactions, particularly as a result of the short heating duration.

Microsecond-scale heating is indeed shorter than typical heating methods employed in diamond-anvil cell experiments or other static high-pressure techniques. Whether equilibrium is reached during such heating depends on whether the reaction rate is faster than the heating duration. Certain reactions occur extremely rapidly; for example, during shock experiments stishovite formation has been observed within a few nanoseconds¹⁰¹.

We found that hydrogen is extremely reactive, and the reactions observed in our experiments occurred spontaneously and very rapidly. Because of its small atomic size, hydrogen diffuses easily within the atomic-scale structure of materials, creating an extensive contact surface between hydrogen and the reacting materials. Once hydrogen diffuses into a material, it can react from within the atomic structure. Moreover, silicate melts react significantly faster than their solid counterparts. In our experiments, all of the involved materials were in a liquid state during the reactions.

One way to assess whether the heating duration was sufficiently long relative to the reaction rate is to extend the heating duration and monitor for further reaction progression. In such tests, we did not observe any clear evidence of additional reaction progression (see runs SCO-10, SCO 13, and SIL-5 in Table 1).

At pressures ≤ 25 GPa, we observed complete release of Si from silicate melts in hydrogen medium during olivine + Fe metal experiments, which are most relevant to the expected compositions of sub-Neptunes. At pressures ≥ 25 GPa, weak diffraction peaks of bridgmanite were observed after melting. For the latter case,

while we cannot entirely rule out the possibility of unreacted starting material, it is also possible that both forward and reverse reactions occurred simultaneously, indicating a potential equilibrium state. Under such conditions, the absence of silicates at lower pressures could reflect hydrogen-saturated conditions, where the reactants were undersaturated relative to hydrogen.

This discussion has been incorporated into lines 152–160 in this revision.

L4 Referee #3

This paper presents a novel and provocative overarching picture: that reduction reactions between an H₂-rich atmosphere and a molten planet can convert a planet from an H-rich atmosphere to a water-rich planet. Hence, a portion of the mass-radius distributions among exoplanets might be explained through a varying hydrogen/water ratio. So, the general interest of the overarching result is likely high and the mechanism is creative.

We appreciate reviewer 3's assessment of this paper, particularly about the impact of our work.

That said, I am not clear that the experiments presented, coupled with dynamic considerations, lead robustly and inconclusively to that overarching picture. The experiments themselves are very challenging—pulsed laser heating of H₂-rich samples to minimize hydrogen diffusion into the anvils, and the authors appear to have done a thoughtful job with their analysis. However, the particular experiments involve (1) abundant metallic iron (a proxy core material), (Mg_{0.9}Fe_{0.1})₂SiO₄ olivine (a decent terrestrial mantle proxy), and what appears to be effectively infinite hydrogen; (2) Fe₂SiO₄-olivine (an endmember) + hydrogen; and (3) SiO₂ (a simple model silicate), iron, and hydrogen.

At the range of pressures and temperatures interrogated (~6-40 GPa, ~2000-4000 K), Reaction 1 generally produces reduced iron silicides, iron hydride and MgO; reaction 2 produces iron hydride, and sometimes iron silicides and sometimes SiO₂; and reaction 3 produces iron silicide, sometimes iron hydride, and SiO₂. Once silicon or divalent iron are reduced, oxygen is released, and Raman spectroscopy shows that water is formed in the samples.

Here's where I have one dilemma: although I recognize that the authors' exoplanet definition of "core" on lines 126-127 includes both the rocky/silicate and iron-rich components, the discussion on lines 124-144 elides from discussing hydrogen reaction with a silicate melt (lines 124-130), to apparently requiring reduced iron metal and molten silicates to be juxtaposed to make the reaction run to its fullest potential. That's in accord with their experiment types (1) and (3). So, for this reaction to run the way it's written, it seems that the iron-alloy core (using core in the terrestrial planet sense) has to be either unformed, or in the process of formation, and fully in reactive equilibrium with the silicate of the magma ocean. That's at the crux of the calculation on lines 145-147, where the model sub-Neptune goes from 0-22 wt% water via reaction #3. The issue here is that the iron alloy core appears to achieve equilibrium with the atmosphere of the planet (and the magma ocean, as well)—and that requires some substantially more extensive explanation. For example, perhaps the standard picture of gas-giant planetary evolution involves accretion of an H₂-envelope around a rocky (rock plus iron) core: versions of this are preserved in recent models and interpreted observations (e.g., Alibert et al., Nature Astronomy, 2, 873, 2018; Weiss and Bottke, AGU Advances, 2, 2021). Given the rapid time-scale of core formation, I'm not sure I see how iron, a magma ocean, and an atmosphere can be fully equilibrated with one another, as Eqn. 3 seems to imply. At a minimum, an extensive discussion of the structural limitations on Eqn. 3 going to completion seems mandated.

We appreciate the reviewer's thoughtful comments, which have inspired us to improve the paper in this revision.

Regarding "... requiring reduced iron metal ...", in this revision we showed that the hydrogen-magma reaction involves two processes, both of which can produce water: the hydride formation reaction (SiH₄; Equation 2) and the Fe-Si alloy formation reaction (Equation 1) (see #3 in section L1.1 for details). The hydride formation process does not involve metallic Fe and therefore does not require Fe metal. In this revision, we added detection of the SiH₄ Raman mode in Extended Data Fig. 3. The formation of SiH₄ has also been documented in recent experiments on heated silicate + hydrogen at lower *P-T* conditions (refs^{25,26,99}). The Raman peak frequency observed in our study is consistent with the Si–H vibration

reported in those studies.

Regarding “... abundant metallic iron ...,” we showed in this revision (lines 92–95; Extended Data Text 9; Supplementary Code 1) that: (1) The total amount of Fe, including Fe metal and Fe²⁺ in olivine, was smaller than the amount of Fe required to form FeSi as the only phase resulting from the complete removal of Si from silicate melt, as observed in our olivine + Fe metal experiments. (2) The Si content in the Fe-Si alloy formed in our experiments was always $y \leq 0.5$ in Fe_{1-y}Si_y (Extended Data Texts 1–3). (3) Not all the Fe in the system was used to form the Fe-Si alloy, as FeH_x alloy was also observed (Fig. 2). Thus, not all the Si released from the silicate melt alloyed with Fe metal; instead, a significant amount of Si reacted directly with hydrogen to form SiH₄.

Regarding “... effectively infinite hydrogen ...,” the newly added Supplementary Code 1 demonstrates that this is not the case (see also lines 147–160 and Extended Data Text 9). We found that 4–6 wt% hydrogen existed in the heated volume. To our knowledge, no experimental technique is currently available to measure the quantities of hydrogen and water directly at high pressure, given the volatility of these components upon pressure quenching. Thus, we estimated these amounts in Supplementary Code 1, including uncertainties and propagating them appropriately. The hydrogen-to-silicate ratio we estimated is comparable to those inferred for sub-Neptunes from their mass-radius relations (lines 147–160).

Regarding “... iron-alloy core (using core in the terrestrial planet sense) has to be either unformed, or in the process of formation, and fully in reactive equilibrium with the silicate of the magma ocean ...,” the underlying assumption in this comment is that Fe-Si alloy formation (Equation 1) is the only process that can produce water. In this revision, we demonstrated that there is a second process for water production in the hydrogen-magma reaction: the SiH₄ formation reaction (Equation 2). Notably, this process does not change the oxidation state of Si, meaning that no reduction is required. As a result, this reaction can occur under more oxidizing conditions, allowing water production even after Fe alloys have fully segregated to form a separate metal layer (lines 202–206).

In this revision, we included additional experimental runs in which hydrogen concentration was significantly reduced by using a 50% Ar + 50% H₂ mixture medium instead of pure H₂ (Extended Data Table 1; Extended Data Text 7). The starting material was fayalite (Fe₂SiO₄), which produces a large amount of water through Fe reduction, quickly raising water activity in the system. Even under these oxidizing conditions, we identified SiH₄, as shown in Extended Data Fig. 3, supporting the continuation of endogenic water production under more oxidizing conditions.

Recent studies have shown that the dynamics of magma oceans in sub-Neptunes can differ significantly due to effective thermal insulation from thick envelopes^{34,35}. In these environments, small metal droplets enriched with light elements may remain suspended in the silicate magma ocean. If this occurs, the Fe-Si alloying reaction identified in our experiments could play an important role over extended periods in planetary evolution (lines 206–210).

My second major issue is also related to the mechanics of how these reactions might proceed, and is encapsulated by the statement on line 127 that “Large amounts of ingassing of hydrogen is possible through physical mixing^{15,16} (sic).” Later in the paper, it is stated (line 211) that the reactions proposed might result in a “fuzzy” boundary between the atmosphere and magma ocean in the “late stage of the conversion” because of H₂O-MgO mixing. But in the early stages, the idea that (say) buoyant MgO might form a proto-crust/septum between the atmosphere and magma ocean, and hence impede reactions that involve atmosphere/magma-ocean equilibration, could also occur. This scenario is pretty much what was published by all of these authors (plus one other) in their Figure 5 in reference 19 (The Planet. Sci. J., 2023), about which they stated “If a FeO-depleted layer forms quickly at the topmost part of the oxide magma ocean where magma reacts with dense hydrogen fluid, it can crystallize and form solid crust, which could shut down reaction 3 and therefore limit the ingassing of H (Figure 5).”

The experiments in ref.²³, cited by the reviewer, were conducted during the early stages of our technical development. At that time, the temperature could not be raised high enough to enable potential reactions between MgO-rich melt and hydrogen. During the revision of this work, in a separate experimental effort, we successfully achieved the melting of MgO in a liquid hydrogen medium, as published in *PNAS* (ref.²⁴).

This work demonstrated that the reaction between MgO and hydrogen produces MgH₂ dissolved in the liquid hydrogen medium, which subsequently generates water. Given the low melting temperatures of the identified phases in that study, the boundary between the hydrogen-rich envelope and the magma is unlikely to act as a barrier. On the contrary, the enhanced solubility between hydrogen and magma would facilitate greater reactions at the CEB (as discussed in Extended Data Text 10).

Similar caveats, beyond a clause in the abstract ('in conjunction with internal dynamics', line 18-19) and a mention that investigations of dynamics are needed (line 188), don't seem to be present here.

In this revision, Prof. A. Vazan joined as a co-author and contributed to the dynamic modeling on how mixing can impact the hydrogen-magma reaction, which is discussed in #2 of section L1.1, lines 181–196, and illustrated in Fig. 4.

My point here would be that there are multiple ways to impede atmosphere-magma ocean equilibration through stratification via the proposed reaction, from formation of an MgO layer, to Fe-depleted magma rising and stratifying, to buoyant H₂-saturated magma stratifying in the near-surface over denser, unsaturated magma. Such effects, which constitute a rather major caveat to the ability of the proposed reactions to dramatically alter a planet's chemistry, also need to have much more extensive discussion.

As the reviewer mentioned, there are factors that can influence the extent of endogenic water production, which we have also discussed in the manuscript (lines 211–236). Additionally, we have addressed most of the factors mentioned in this question (such as the “formation of an MgO layer” and “Fe-depleted magma rising and stratifying”) in our responses to the reviewer’s previous questions.

Regarding the reviewer’s hypothesis of “... H₂-saturated magma stratifying in the near-surface ...,” we show that the intense reaction continues at least up to 50 GPa. Furthermore, our recent work published in *PNAS* (ref.²⁴) demonstrated that pressure enhances the mutual solubility between hydrogen and MgO melt, contrary to the demixing required for the stratification hypothesized by the reviewer. In this scenario, where the magma ocean is at very high temperatures, the core could act as a sink for hydrogen, promoting the reaction as greater depths remain hydrogen-undersaturated.

Thus, mounting evidence supports more intense physical and chemical interactions between hydrogen and the core. In fact, no clear factors have been identified that would completely shut down endogenic water production.

My net take on the paper is that the experiments are difficult, timely and provocative. However, the grand scenario proposed in the paper, and presented as a straightforward means of moving from H₂-dominated to H₂O-abundant sub-Neptune planets, has some quite major mechanistic difficulties, which are scarcely, if at all, described. In short, it looks like there's a bunch of ways that this scenario might well not work, or might work minimally. These include early silicate-iron differentiation (apparently) and interaction-induced stratification at the top of the magma-ocean. That neglect of major caveats doesn't give the full story to the interested communities. As such, I believe the paper needs to be largely reworked before it can be considered for publication.

We appreciate the reviewer’s assessment of the strengths and weaknesses of this work. In this revision, we have addressed all the factors raised by the reviewer that could potentially negatively impact endogenic water production. We also provide new lines of evidence supporting the production of significant amounts of endogenic water through hydrogen-magma reactions in hydrogen-rich sub-Neptunes.

We acknowledge and have considered that the extent of water produced through endogenic processes can vary among sub-Neptunes due to the factors discussed in the manuscript, as outlined above. However, what is robustly established in this work is that pressure enhances endogenic water production, and that larger rocky bodies, such as super-Earths converted from sub-Neptunes, likely contain much more water than smaller rocky bodies, such as Earth.

References

97. Lin, J.-F. *et al.* High pressure-temperature Raman measurements of H₂O melting to 22 GPa and 900 K. *The J. Chem. Phys.* **121**, 8423–8427, DOI: [10.1063/1.1784438](https://doi.org/10.1063/1.1784438) (2004).
98. Schwager, B., Chudinovskikh, L., Gavriluk, A. & Boehler, R. Melting curve of H₂O to 90 GPa measured in a laser-heated diamond cell. *J. Physics: Condens. Matter* **16**, S1177–S1179, DOI: [10.1088/0953-8984/16/14/028](https://doi.org/10.1088/0953-8984/16/14/028) (2004).
99. Shinozaki, A. *et al.* Preferential dissolution of SiO₂ from enstatite to H₂ fluid under high pressure and temperature. *Phys. Chem. Miner.* **43**, 277–285, DOI: [10.1007/s00269-015-0792-3](https://doi.org/10.1007/s00269-015-0792-3) (2016).
100. Yagi, T. & Hishinuma, T. Iron hydride formed by the reaction of iron, silicate, and water: Implications for the light element of the Earth's core. *Geophys. Res. Lett.* **22**, 1933–1936 (1995).
101. Tracy, S. J., Turneure, S. J. & Duffy, T. S. *In Situ* X-ray diffraction of shock-compressed fused silica. *Phys. Rev. Lett.* **120**, 135702, DOI: [10.1103/PhysRevLett.120.135702](https://doi.org/10.1103/PhysRevLett.120.135702) (2018).

Response to Reviewers

L1 Referee #1

As I indicated in my initial review, this is an interesting paper that is presenting a new idea about how planets might have gotten their water. The paper is based on some innovative and challenging experimental work that investigates the environment in the interior of sub-Neptunes and super-Earths. They show that it is possible to generate water inside of a nominally anhydrous planets through reduction of SiO₂ to Si⁴⁺ with the resulting oxygen combining with hydrogen to form water. This idea is new to the planetary science community and, if correct, would significantly change the way we view planet formation and how we think the planets got their water.

We are grateful that the reviewer finds our paper interesting and acknowledges the innovative and technically challenging nature of our experimental work. We also appreciate the reviewer's recognition of the broader implications of our findings for planetary science—particularly regarding the origin of water in sub-Neptunes and super-Earths. As the reviewer notes, the possibility of forming water internally within nominally dry planets through chemical reactions presents a new mechanism that could reshape our understanding of planetary formation and volatile delivery. We agree that this concept may have far-reaching consequences not only for exoplanets but also for water distribution among solar system bodies.

The new version of the paper is more focused to address the primary contribution of this work. The text is easier to read and better organized than the original.

We are pleased that the reviewer finds the revised manuscript to be clearer, more focused and easier to read.

The authors took the reviewer's comments very seriously and provided good responses to each of the comments. The authors addressed each of the comment in detail. Although I am not an expert in this experimental area, I found the discussions with the reviewers to be productive and the authors took those comments seriously. I learned a lot from the responses to the reviewers.

We thank the reviewer for their thoughtful engagement and kind words. We are glad that our detailed responses helped clarify aspects of our work and contributed to productive discussions. We also agree that the reviewers' questions played a key role in improving the quality and clarity of the manuscript.

The current paper does not answer all of the questions from the reviewers. But it is not realistic to expect complete answers. This is a new field and this project is at the state of the art. I think the authors have done a good job of explaining the experimental techniques and interpreting their data. They point the way for the next generation of experiments (which will be difficult, but not impossible). The work is mature enough to introduce it to the world and have the community study the new ideas.

We agree that this paper marks an important starting point for new directions in understanding the origin of water in planetary settings, while also raising some important questions to be addressed in future studies. We hope our work will stimulate further experimental efforts to investigate materials under the extreme and diverse conditions relevant to exoplanets. The importance of laboratory investigations in exoplanet science has been underscored in several recent decadal surveys and white papers^{104, 105}. However, progress has been limited, in part due to the challenging nature of exoplanetary conditions and gaps in communication between the modeling, observational, and experimental communities.

We hope this study serves as a valuable example of how experimental approaches can contribute to bridging these gaps and advancing our understanding of planetary processes beyond the solar system.

My suggestions at this point are basically cosmetic.

For example, I don't understand the page numbers and section numbers. Figures 2 and 3 are easier to understand than in the first version. Figure 4 caption, last sentence – add an “s” to “show”. (diagram shows)

We made the suggested correction.

After taking a final look at the entire paper, I think the paper will be ready to publish.

We are pleased that the reviewer finds the revised manuscript “ready to publish”. We also appreciate the earlier recognition of the “maturity” of our work and the constructive feedback throughout the review process, which helped us improve the paper.

L2 Referee #3

As with the prior version on the paper, this paper presents some quite provocative observations about reactions between silicates (\pm iron) and hydrogen at high pressure and temperature conditions, with possible relevance to endogenous water production in hydrogen-mantled planets. The discussion of the data is improved in this version, and (perhaps as an illustration of how difficult to monitor, or likely to evolve in their chemical complexity these reactions are) a new and possibly critically important ingredient/reaction product is introduced in this version of the paper relative to the previous manuscript version: SiH₄. Reactions involving this component do not require equilibration with iron (in my original review, I was skeptical that reactions involving metallic iron would inevitably occur between iron and the overlying H₂ layer, due to likely rapid core-mantle segregation processes). So, the introduction of this new component/reaction allows water-forming reactions with a silicate layer that need not contain iron metal. That's probably more plausible than the initial scenario.

We are encouraged that the reviewer finds the revised scenario more plausible, especially with the inclusion of SiH₄ as a reaction product. We agree that the formation of SiH₄ opens a new pathway for endogenic water production that does not require the presence of metallic iron, addressing a key concern raised in the earlier review.

In fact, the reviewer's insightful critique was a major motivation for us to revisit and re-analyze the data, which led to the identification of this important reaction pathway. We appreciate the opportunity to improve the paper through this constructive feedback.

So, on the positive side, the inclusion of lines 181-201 (and the two paragraphs following) do improve the paper. The accompanying Figure 4 is also a positive addition, but only addresses the possible evolution of an H₂O-rich septum between the silicate magma region and H₂/He atmosphere (which lines 181-183 note would markedly impede the proposed H₂O-forming reactions).

We appreciate the reviewer's recognition of the improvements in this version, particularly the addition of lines 181–201 and the accompanying discussion. We agree that incorporating both dynamic and thermodynamic considerations is crucial for understanding the implications of the reported chemical reactions. The inclusion of Figure 4 and the associated discussion was motivated by this important point raised by Reviewer #3, and we are glad that the reviewer finds it a valuable addition to the manuscript.

That said, my major concerns with the paper remain dynamical in nature. In my view, the new result –silicon hydride production, and to a lesser extent MgH₂, as described on the top of page 14 of the rebuttal but minimally mentioned (line 52) in the main manuscript, may well produce a supplementary suite of density stratification issues that are not discussed in the current manuscript. Any possible stratification associated with the new reaction products (e.g., a SiH₄/MgH₂-hydride containing H₂O/H₂ layer) is not mentioned in the manuscript.

We agree with the reviewer that the dynamical implications of hydride formation—particularly its impact on density stratification—are important considerations. As noted, our focus in the manuscript is on SiH₄ because it is the dominant hydride observed in our experiments. Based on our estimates, SiH₄ can form up

to 6–23(5) mol%, as now discussed in lines 205–227 of the revised manuscript.

In contrast, the formation of MgH_2 is expected to be far more limited. As we note in lines 53–55 and Extended Data Text 11, MgH_2 forms only at very high temperatures (above 3000 K)²⁴, which were not fully reached in our experiments. Consequently, we did not observe any clear signature of MgH_2 , likely due to its limited abundance and the temperature range explored.

Moreover, MgH_2 formation is expected to occur during the early, high-temperature stage of a sub-Neptune's evolution, when the system is likely experiencing even more vigorous convection. At that time, the envelope would not yet contain a significant amount of dense reaction products such as H_2O , allowing MgH_2 to be efficiently mixed into the convecting envelope. This is consistent with the early-stage mixing shown in one of the higher-temperature panels of Fig. 4.

Nonetheless, we acknowledge the importance of discussing possible density stratification effects due to SiH_4 , which is discussed in our answer for the next question, including how we revised our manuscript for this point.

Indeed, while H₂-H₂O miscibility is proposed to enhance mixing between layers (lines 193-197) through buoyancy forces, the negative buoyancy forces and possible stratification that are likely to be associated with possible SiH₄ and MgH₂ mixing/solubility in either the H₂O-rich layer (plus/minus H₂) seem to be undiscussed. I would guess that this could generate more severe density stratifications than are probed in Figure 4, and hence layers that could be stable for longer periods/higher temperatures. So, while I appreciate the authors' much-improved discussion of dynamic effects, I do think that the statement near the base of p. 14 in the rebuttal ("In fact, no clear factors have been identified that would completely shut down endogenic water production") and the calculations of Figure 4 may not be fully reflective of the enhanced chemical complexity that the authors have newly introduced to the water-forming reactions in this version of the paper. My concern here is simple: given the production of Si and Mg-hydrides, Figure 4 may represent a near-best-case scenario for keeping these reactions going in planetary interiors.

We agree that hydride formation may influence density stratification and therefore the long-term dynamics of these systems.

Mixing between SiH_4 and H_2 Although data are limited, experimental studies indicate that SiH_4 and H_2 are completely miscible up to 6 GPa in the liquid state at 300 K (ref.⁸⁵), and they even become miscible in solid form at high pressure⁸⁶. Given the high temperatures (2000–4000 K) relevant to our study and sub-Neptunes' CEB conditions, and the low melting point of SiH_4 , this suggests that SiH_4 should remain in the liquid state and be miscible with H_2 throughout the relevant pressure range.

We want to make clear that the mixing discussed here is through *miscibility of two or more fluids resulting in a single fluid*, while the mixing included in our model is convectational mixing of separate fluids, mainly because there is currently insufficient data to quantitatively describe the miscibility and density of SiH_4 – H_2 mixtures. This miscibility is crucial: even if a denser fluid layer enriched in H_2O and SiH_4 forms at the base of the envelope, H_2 will remain present in the layer of dense fluid through miscibility (in other words, the dense fluid still contains H_2) and can continue to interact with the underlying magma, sustaining water-forming reactions. We also note that during the review of this manuscript, a new study, which demonstrated complete miscibility of H_2 and H_2O at relevant P – T conditions, forming a single fluid phase³¹, is now peer reviewed and published.

Density of SiH_4 and MgH_2 The reviewer raises a valid point about the potential role of hydride densities in driving stratification. While the molar mass of SiH_4 is higher than H_2O , its density is actually lower. As shown in Fig. L1a, the density of solid SiH_4 at high pressure and 300 K is approximately 20–30% lower

than that of H_2O , primarily due to its significantly larger molar volume. This arises from several factors: (1) SiH_4 is a symmetric, nonpolar molecule, leading to greater intermolecular repulsion and lower packing efficiency; (2) hydrogen in SiH_4 and MgH_2 exists as H^- , which has much larger ionic radius (1.34 Å) than H^+ (0.18 Å) in H_2O ; and (3) H_2O 's strong hydrogen bonding results in a more compact structure.

Although fluid-phase densities remain unmeasured at relevant P - T conditions, it is reasonable to expect a similar trend in relative density differences. Based on our estimates of SiH_4 concentration (6–23 mol%, now included in lines 205–227), the net effect would likely be a *reduction* in density of the H_2O -rich layer, not an increase. Therefore, the presence of SiH_4 may *mitigate* rather than exacerbate stratification.

A similar conclusion applies to MgH_2 . Although its abundance is limited—only expected to form above ~ 3000 K and not observed in our experiments—it also appears to be less dense than H_2O based on available solid-state data (Fig. L1b). Moreover, MgH_2 is likely to form during early, high-temperature phases when vigorous convection is dominant and before substantial quantities of denser reaction products like H_2O accumulate. This would enable efficient convective mixing, as illustrated in the early-stage panel of Fig. 4.

For these reasons, we respectfully suggest that Fig. 4 reflects a relatively conservative scenario rather than a “near-best-case one”. Another factor to consider is that our model did not include potential *ingassing* of H_2O and SiH_4 into the underlying magma ocean, which would further reduce the concentrations of heavy species in the envelope and thus suppress stratification (as discussed in lines 196–197).

We have incorporated these clarifications in the revised manuscript (lines 205–227) and Expanded Data Text 12. We thank the reviewer for highlighting this issue, which has led us to further strengthen our discussion of the interplay between chemistry and planetary dynamics.

Fig. L1. Comparison of (a) density and (b) molar volume of solid H_2O , SiH_4 , and MgH_2 . To our knowledge, no density measurement exists for SiH_4 and MgH_2 fluids at the pressure-temperature conditions of this study. However, the comparison of solid phases at 300 K and high pressure provides insights on the density different among fluid phases. The equations of state of H_2O (ice VII) and SiH_4 are from ref.⁸⁷ and ref.⁸⁸, respectively. The equations of state of α - MgH_2 (a lower-pressure polymorph) and ϵ - MgH_2 (a higher-pressure polymorph) are from ref.⁸⁹ and ref.⁹⁰, respectively.

My net take is thus that there still need to be some important caveats (and maybe, should the authors choose, additional

calculations) introduced in conjunction with statements about the overarching ability of these reactions to, as the title indicates, facilitate "Building a Wet Planet from Dry Materials...". My sense is that the introduction of new reaction products without considering or discussing their effect on the dynamics of the system does motivate revisions to the manuscript that treat their role in generating possible stratification that might occur in evolving planets, and in impeding (or, if there's something creative that I've missed, enhancing) water-generating reactions that might occur between layers/chemical components in their interiors.

As discussed above, our analysis suggests that the formation of hydrides such as SiH_4 and MgH_2 may in fact mitigate, rather than exacerbate, potential stratification at the base of the envelope. We have incorporated this discussion into the revised manuscript (lines 205–227) and Extended Data Text 12.

We also agree with the reviewer that statements about the overarching implications of these reactions must include appropriate caveats, especially given the complexity and variability of planetary evolution scenarios. In response, we have revised the abstract to reflect the potential range in water production outcomes and the role of dynamic processes in modulating these reactions (lines 19–22), as well as additional caveats and qualifying statements throughout the manuscript. We thank the reviewer for this valuable suggestion, which helped us improve the clarity and balance of our conclusions.

References

104. Committee on the Planetary Science and Astrobiology Decadal Survey, Space Studies Board, Division on Engineering and Physical Sciences & National Academies of Sciences, Engineering, and Medicine. *Origins, Worlds, and Life: A Decadal Strategy for Planetary Science and Astrobiology 2023-2032* (National Academies Press, Washington, D.C., 2023).
105. Committee on Exoplanet Science Strategy, Space Studies Board, Board on Physics and Astronomy, Division on Engineering and Physical Sciences & National Academies of Sciences, Engineering, and Medicine. *Exoplanet Science Strategy* (National Academies Press, Washington, D.C., 2018).